# An ensemble-based coupled reanalysis of the climate from 1860 to the present (CoRea1860+)

Yiguo Wang[1], François Counillon[1], Lea Svendsen[2], Ping-Gin Chiu[2], Noel Keenlyside[2], Patrick Laloyaux[3], Mariko Koseki[2], and Eric de Boisseson[3]

[1]Nansen Environmental and Remote Sensing Center and Bjerknes Centre for Climate Research, Bergen, Norway
[2]Geophysical Institute, University of Bergen, and Bjerknes Centre for Climate Research, Bergen, Norway
[3]European Centre for Medium-Range Weather Forecasts, Reading, UK

**Correspondence:** Yiguo Wang (yiguo.wang@nersc.no)

**Abstract.** Climate reanalyses are essential for studying climate variability, understanding climate processes, and initializing climate predictions. We present CoRea1860+ (Wang and Counillon, 2025, https://doi.org/10.11582/2025.00009), a 30-member coupled reanalysis spanning from 1860 to the present, produced using the Norwegian Climate Prediction Model (NorCPM) and assimilating sea surface temperature (SST) observations. NorCPM combines the Norwegian Earth System Model with the ensemble Kalman filter data assimilation method. SST, available throughout the entire period, serves as the primary source of instrumental oceanic measurements prior to the 1950s. CoRea1860+ belongs to the category of sparse-input reanalyses, designed to minimize artifacts arising from changes in the observation network over time. By exclusively assimilating oceanic data, this reanalysis offers valuable insights into the ocean's role in driving climate system variability, including its influence on the atmosphere and sea ice. This study first describes the numerical model, SST dataset, and assimilation implementation used to produce CoRea1860+. It then provides a comprehensive evaluation of the reanalysis across four key aspects: reliability, ocean variability, sea ice variability, and atmospheric variability, benchmarked against more than ten independent reanalyses and observational datasets. Overall, CoRea1860+ demonstrates strong reliability, particularly in observation-rich periods, and provides a reasonable representation of climate variability. It successfully captures key features such as multidecadal variability and long-term trends in ocean heat content, the Atlantic meridional overturning circulation, and sea ice variability in both hemispheres. Furthermore, CoRea1860+ agrees with the reference atmospheric datasets to some extent for surface air temperature, precipitation, sea level pressure, and 500 hPa geopotential height, especially in the tropics where air-sea interactions are most pronounced.

## 1 Introduction

Long observational records are indispensable for studying climate variability and gaining a deeper understanding of climate change over the past century, particularly in the context of anthropogenic forcings. Many efforts to improve the availability and quality of observational data (i.e., observations) have been ongoing for decades, with significant advancements achieved through data rescue initiatives recovering historical observations and modern observing platforms, such as satellites and Argo floats, which provide near-global coverage of the climate variables. However, despite these advancements, observations are of-

ten sparse in both time and space, particularly in remote or inaccessible regions such as the interior ocean, polar areas, and parts

of the atmosphere. In addition, the existing observing systems observe or monitor only a few climate system variables, such as air temperature, precipitation, ocean temperature, and sea ice concentration. Thus, based solely on observations, constructing a seamless and comprehensive record of the climate system remains a big challenge.

Retrospective analyses (i.e., reanalyses, Kalnay et al., 1996; Wang et al., 2023) are a comprehensive four-dimensional reconstruction of the historical climate system achieved by combining observational data (i.e., observations) with a numerical

physical model through data assimilation (DA, Evensen, 2003; Carrassi et al., 2018; Penny et al., 2017). This process leverages the strengths of both observational datasets and model simulations, with DA enabling observational information to be propagated across time and space to fill gaps in unobserved areas and variables. This is accomplished under the physical constraints imposed by the numerical model, ensuring both dynamical consistency and accuracy of the reconstructed climate state. Reanalysis serves as a vital tool for understanding climate variability and assessing climate change over time. Beyond these

applications, it offers valuable insights into large-scale climate teleconnections, such as Eurasian cooling (Outten et al., 2023) and Arctic warming (Ding et al., 2014; Cai et al., 2021), where it aids in revealing complex relationships between different components of the climate system. Moreover, reanalysis products play a critical role in initializing climate predictions, ranging from short-term forecasts spanning a few weeks to decadal or even longer-term projections (e.g., Balmaseda et al., 2009; Brune et al., 2015; Boer et al., 2016; Wang et al., 2019; Bethke et al., 2021; O'Kane et al., 2021; Polkova et al., 2023).

Reanalyses can broadly be categorized into uncoupled and coupled reanalyses, depending on the numerical model used in their production. Uncoupled reanalyses are generated using numerical models that simulate only certain components of the Earth system. For instance, atmosphere-land reanalyses (e.g., ERA5, ERA-20C, 20CRv3, Hersbach et al., 2020; Poli et al., 2016; Slivinski et al., 2021) are produced with atmosphere-land coupled models that are forced by prescribed surface data, such as sea surface temperature (SST) and sea ice concentration. Similarly, ocean reanalyses (e.g., ORAS5, ORA-20C, SODA2.2.4,

CHOR, Zuo et al., 2019; de Boisséson et al., 2018; Giese et al., 2016; Yang et al., 2017) use ocean-sea ice coupled models, or sometimes standalone ocean models, driven by prescribed atmospheric data. These uncoupled reanalyses are generally very accurate in their specific domain of the climate system, providing reliable reconstructions of their respective components.

On the other hand, coupled reanalyses are generated by numerical models that simulate coupled atmosphere-ocean dynamics, enabling them to account for the complex interactions between different components of the climate system. Examples of

coupled reanalyses include CERA-20C (Laloyaux et al., 2018), the NCEP Climate Forecast System Reanalysis (Saha et al., 2010), CAFE60V1 (O'Kane et al., 2021), Multivariate Ocean Variational Estimation System–Coupled Version Reanalysis (MOVE-C RA, Fujii et al., 2009), and reanalyses produced with the Norwegian Climate Prediction Model (NorCPM, Counillon et al., 2016; Bethke et al., 2021). The atmosphere-ocean coupling in these reanalyses allows them to capture critical coupled processes that are absent in uncoupled reanalyses. For instance, CERA-20C effectively captures the westward propagation of

tropical instability waves, which play a crucial role in the interannual variability and predictability of the El Niño-Southern Oscillation (ENSO). By contrast, the uncoupled reanalysis ERA-20C can not represent these coupled dynamics because they do not simulate interactions between the atmosphere and ocean (Laloyaux et al., 2018). This distinction highlights the enhanced

capability of coupled reanalyses to provide a more comprehensive and physically consistent representation of the climate system.

Due to the limited availability of observations during the first half of the $20^{th}$ century, most coupled reanalyses (Counillon et al., 2016; Brune et al., 2015; O'Kane et al., 2021) have been produced for periods beginning in the 1950s or later. However, this relatively short period is insufficient for studying the evolution of slow modes of climate variability, such as the Atlantic Multidecadal Variability (AMV, Omrani et al., 2022), the Pacific Decadal Variability (Newman et al., 2016), tropical basin interactions (Cai et al., 2019; Wang, 2019), and their long-term influence on the climate system. To date, CERA-20C (Laloyaux

et al., 2018) remains the only coupled reanalysis that spans the entirety of the $20^{th}$ century. However, its production process involved the parallel generation of 14 ten-year production streams (initialized from the uncoupled ERA-20C and ORA-20C reanalyses) from which the first two years are discarded to produce the final climate reconstruction for the period 1901–2010, leading to discontinuities in the ocean variables (see Figure 10 in Laloyaux et al., 2018). Furthermore, CERA-20C assimilated ocean subsurface temperature and salinity profiles whose global coverage was notably poor in the first half of the $20^{th}$ century

but significantly improved in the recent two decades. The evolution of subsurface observation networks likely yields discontinuities or inconsistencies in reanalysis. One way to mitigate these discontinuities is to produce the reanalysis as a single continuous stream while excluding observations that are not consistently available throughout the entire period. For example, 20CRv3 (Slivinski et al., 2019) addressed this issue by creating a $20^{th}$century atmosphere reanalysis that only assimilated sea level pressure data while prescribing SST at the surface—an approach known as sparse-input reanalysis.

One key challenge in coupled reanalysis is effectively propagating observational information across different climate system components during DA, a process known as coupled DA (Penny et al., 2017). Coupled DA can be classified into two types: semi-coupled DA and fully-coupled DA. In semi-coupled DA, observations are assimilated into their respective components (Counillon et al., 2016; Kimmritz et al., 2019). For instance, when assimilating atmospheric and oceanic data in semi-coupled DA, the atmospheric data are only used to update the atmosphere component and the oceanic data are solely used to update the

ocean component. Semi-coupled DA still allows for constraints on the other components through the model's coupling of the components, making it valuable for studying specific component interactions within the climate system. In fully-coupled DA, all components are directly constrained by observations (Fujii et al., 2009; Laloyaux et al., 2016). Again, for example, when assimilating atmospheric and oceanic data, the atmospheric data are used to update not only the atmosphere component but also the ocean component and the oceanic data are used to update both the ocean and atmosphere components. Although fully-

coupled DA theoretically outperforms semi-coupled DA by incorporating more observations (Penny et al., 2017), practical challenges arise due to the differing spatial and temporal scales of the components. In NorCPM, for example, attempts to constrain the atmosphere component have led to a degradation in the performance of the ocean component constraint (Garcia-Oliva et al., 2024). To address these challenges, our approach focuses on constraining the ocean component of the climate system, allowing CoRea1860+ to serve as a coupled climate reanalysis suited for studying the ocean's role as a driver of

climate interactions.

Another key challenge in coupled reanalysis is managing model bias, as coupled models often exhibit significant biases due to inherent model deficiencies (Richter, 2015). The full-field assimilation directly uses the actual value of observations and

corrects both the model state and variability (de Boisséson et al., 2018; Slivinski et al., 2021). However, this approach can lead to persistent model drift, where the coupled system repeatedly moves away from observations toward its biased state (Carrassi et al., 2014; Weber et al., 2015). This drift introduces inconsistencies in the coupled reanalysis, particularly in unobserved variables and regions such as the deep ocean when observational data are sparse. An alternative approach, anomaly-field assimilation, assimilates observed climate anomalies rather than absolute values, maintaining the model state closer to its own attractor and reducing drift (Carrassi et al., 2014; Weber et al., 2015). This method has been widely adopted in the climate prediction community to address prediction drift (e.g., Magnusson et al., 2013; Smith et al., 2013; Polkova et al., 2023; Xiu et al., 2025). NorCPM, as a climate prediction system, implements anomaly-field assimilation and has demonstrated stable performance (Counillon et al., 2016; Kimmritz et al., 2019; Wang et al., 2019; Bethke et al., 2021; Xiu et al., 2025).

In this study, we aim to present an ensemble-based coupled reanalysis of the climate spanning from 1860 to the present (CoRea1860+, Wang and Counillon, 2025, https://doi.org/10.11582/2025.00009). This reanalysis, produced using 30 ensemble members within the fully coupled climate model NorCPM (Section 2), is generated in a single continuous stream to ensure consistency, which is essential for investigating climate variability on long timescales. Our approach focuses exclusively on assimilating SST data (Section 2), omitting ocean subsurface observations that have only become widely available over the past two decades, and observations in the other components (e.g., atmosphere, land, and sea ice). Furthermore, the reanalysis is produced by anomaly-field assimilation, ensuring that the assimilation keeps the model close to its attractor and thus limits model drift (i.e., large inconsistencies) during model integration. These selective assimilation strategies further enhance the continuity and consistency of the reanalysis product, making it a robust resource for studying long-term climate variability and slow modes of the climate system.

The following section provides an overview of the numerical model and dataset used to produce the CoRea1860+ reanalysis. Section 3 introduces the datasets and metrics employed for the evaluation of CoRea1860+. Section 4 assesses the reanalysis in terms of its reliability, ocean variability, sea ice variability, and atmospheric variability. Finally, Section 5 discusses the findings, highlights related caveats, and concludes the study.

## 2 Norwegian Climate Prediction Model

NorCPM is a physics-based numerical model (scientific software) developed for performing climate reconstructions (Counillon et al., 2016; Wang et al., 2022) and predictions on different timescales (Wang et al., 2019; Bethke et al., 2021; Nair et al., 2024; Xiu et al., 2025). It combines the Norwegian Earth System Model (NorESM) with the ensemble Kalman filter (EnKF) and has 30 ensemble members (i.e., realizations). This section will present the NorESM version used, the assimilated SST dataset, and DA implementation.

## 2.1 Norwegian Earth System Model

NorESM is a state-of-the-art Earth system model that has contributed to the Coupled Model Intercomparison Project phase 5 (CMIP5, Taylor et al., 2012) and phase 6 (CMIP6, Eyring et al., 2016), which have provided input to assessment reports of the Intergovernmental Panel on Climate Change (e.g., Stocker et al., 2013; IPCC, 2023).

The version of NorESM used in this study is the medium-resolution NorESM1-ME (Bentsen et al., 2013), which contributed to CMIP5. It is based on the Community Earth System Model version 1.0.3 (CESM1, Hurrell et al., 2013), a successor to the Community Climate System Model version 4 (Gent et al., 2011). The land component is the Community Land Model version 4 (CLM4, Oleson et al., 2010; Lawrence et al., 2011). The sea ice component is the Los Alamos Sea Ice Model version 4 (CICE4, Gent et al., 2011; Holland et al., 2012), which includes five ice thickness categories and employs the elastic–viscous–plastic rheology. Both of these components are used in their original form as adopted from CESM1. The atmosphere component is an updated version of the Community Atmosphere Model version 4 (CAM4, Neale et al., 2010), featuring a prognostic aerosol life-cycle formulation that replaces the original prescribed aerosol approach. This version incorporates emissions and new aerosol-cloud interaction schemes (Kirkevåg et al., 2013). The ocean component is the Bergen Layered Ocean Model (BLOM, Bentsen et al., 2013), a revised version of the Miami Isopycnic Coordinate Ocean Model (Bleck et al., 1992), designed to minimize spurious diapycnal mixing and improve the conservation of water properties. Finally, the model employs the version 7 coupler (Craig et al., 2012), which seamlessly integrates the different components of NorESM.

The atmosphere and land components of the used version of NorESM share a horizontal resolution of 1.9° in latitude and 2.5° in longitude. The atmosphere component consists of 26 hybrid sigma–pressure levels, extending up to 3 hPa. The ocean and sea ice components have a horizontal resolution of approximately 1°. The ocean component includes 51 isopycnic layers, along with a bulk mixed layer represented by two layers with time-evolving thicknesses and densities.

The NorESM used in this study is forced by CMIP5 historical forcings before 2005, and the Representative Concentration Pathway 8.5 forcings after 2005 (Bentsen et al., 2013). The CMIP5 historical forcings from 1850 to 2005 are based on observational variations in solar radiation (Lean et al., 2005; Wang et al., 2005), volcanic sulphate aerosol concentration (Ammann et al., 2003), Greenhouse gas concentration (Lamarque et al., 2010), aerosol emission (Lamarque et al., 2010), and land use (Hurtt et al., 2009). We have extensive experience with this NorESM version (e.g., Counillon et al., 2014, 2016). Most of our DA system has been specifically tuned for this setup (e.g., Wang et al., 2016, 2017; Kimmritz et al., 2018; Wang et al., 2022). While CMIP6 forcings represent the latest update, their implementation in NorCPM has introduced issues. For instance, Passos et al. (2023) have reported that the CMIP6-forced version of NorCPM suffers from artificial bugs in the land use updates, leading to unrealistic land–cryosphere cooling trends. These artefacts result in slightly larger global mean biases and RMSEs compared to the CMIP5-forced version. Further details can be found in the Supplement of Bethke et al. (2021). Therefore, we opted for producing the reanalysis with the robustly tested version of the system that uses the CMIP5 forcing configuration.

The initial conditions of the reanalysis featuring 30 ensemble members are taken from a 30-member historical simulation of NorESM that was integrated from 1850 to 1860 using the CMIP5 historical forcings. The initial ensemble of the historical simulation of NorESM in 1850 is sampled from a stable and long preindustrial forcing run of NorESM.

## 2.2 Assimilated dataset

Only SST data are assimilated to produce the reanalysis CoRea1860+. From 1860 to 2010, the monthly SST data are taken from the Hadley Centre Sea Ice and Sea Surface Temperature dataset version 2.1 (HadISST2.1, Rayner et al., 2003). Since 2011, the monthly SST data from the Optimum Interpolation SST version 2 (OISSTV2, Reynolds et al., 2002) are assimilated, because HadISST2.1 is only available until 2010.

HadISST2.1 is available over 1850–2010 with 1° resolution and has ten realizations of monthly gridded SST. Its data sources are in situ observations from the International Comprehensive Ocean-Atmosphere Data Set (ICOADS) and the Met Office observational database and SST retrievals from AVHRR Pathfinder data and the ATSR2 and AATSR METEO products. The standard deviation between the ten realizations, which varies with time and space, is designed to reflect its uncertainties (a key quantity for DA). HadISST2.1 has been used in several long reanalyses (e.g., CERA-20C and 20CRv3, Laloyaux et al., 2018; Slivinski et al., 2019) (Section 3).

OISSTV2 is a spatially gridded SST product since 1981 with 1° resolution and created by interpolating and extrapolating data from satellites and in situ platforms (e.g., ships and buoys) with Optimum Interpolation (OI). Its monthly SST data are used in producing the reanalysis CoRea1860+. Since the observation error variance of the monthly SST data is not provided by the producer, it is estimated as the harmonic mean of weekly error variances that are available in OISSTV2.

The SST data in the regions covered by sea ice are not assimilated. These regions are identified using the sea ice mask in HadISST2.1 or OISSTV2.

## 2.3 Assimilation implementation

The EnKF (Evensen, 2003) is an advanced, ensemble-based, and recursive DA method. One key advantage of the EnKF is its probabilistic nature, which enables the quantification of model uncertainty through Monte Carlo ensembles. The EnKF provides multivariate updates, allowing information to be transferred from observed variables to unobserved variables based on their covariances. Additionally, the covariances are computed from the ensemble of the evolving state of the climate system. This capability is particularly crucial for capturing climate regime shifts (Counillon et al., 2016). We utilize in this study a deterministic variant of the EnKF (DEnKF, Sakov and Oke, 2008), which updates the ensemble perturbations by employing an expansion of the expected correction to the background covariances. This approach provides an approximate yet deterministic alternative to the traditional stochastic EnKF, offering improved performance, especially when working with small ensemble sizes (Sakov and Oke, 2008).

Our reanalysis system employs 30 ensemble members, which is relatively small compared to the system's dimensionality. To mitigate spurious correlations arising from sampling errors, we implement the localization technique developed by Houtekamer and Mitchell (2001). Specifically, the local analysis framework (Evensen, 2009) is utilized, where DA is conducted at each horizontal grid cell. Observations within the localization radius of the target grid cell are used to update the model state at this grid cell. To ensure smooth increments and avoid discontinuities at the boundaries of the local domain, we taper observation error variances using the reciprocal of the Gaspari and Cohn function (Gaspari and Cohn, 1999), a function of the distance

between the observation location and the target grid cell. The localization radius in NorCPM varies as a bimodal Gaussian function of latitude (Wang et al., 2017). At the equator, where covariances are anisotropic, it has a local minimum of 1500 km. It reaches a maximum of 2300 km in the middle latitudes and exhibits another minimum in the high latitudes, where the Rossby radius is smaller. This adaptive localization radius ensures that the DA system captures spatial variability effectively across different regions of the globe. The local analysis approach also reduces the dimensionality of the problem, making the EnKF process more computationally efficient.

We perform anomaly-field assimilation in which the climatology of the observations is replaced by the model climatology calculated from the ensemble mean of the NorESM 30-member historical simulation (without assimilation). Referring to Bethke et al. (2021), we define 1950-2009 when monthly SST estimates are relatively accurate for estimating monthly climatology as the climatology reference period to avoid subsampling the internal variability of the climate. The anomaly-field assimilation helps to reduce the model drift during the monthly model integration (Carrassi et al., 2014; Weber et al., 2015; Bethke et al., 2021).

We assimilate monthly SST data (Section 2.2) to update the instantaneous model state at 00:00 UTC on the 15th of each month. The innovation in the EnKF compares an instantaneous model snapshot with monthly-averaged observations (Counillon et al., 2016; Bethke et al., 2021). Billeau et al. (2016) investigated an alternative approach, where the innovation used monthly-averaged model output, and the instantaneous model state was updated at the end of the month. However, they found that this method resulted in a poorer reanalysis performance compared to the former approach.

All ocean state variables (e.g., temperature, salinity, velocity, and layer thickness) are updated in isopycnal coordinates through the assimilation of SST data. Previous studies (e.g., Gavart and Mey, 1997; Counillon et al., 2016) have shown that performing assimilation in isopycnal coordinates efficiently utilizes surface observations. One challenge in this process is that layer thickness, an ocean state variable in BLOM, is inherently non-negative. However, due to the Gaussian assumptions of the EnKF, negative values may occasionally arise. To address this, we apply the aggregation approach proposed by Wang et al. (2016), ensuring that heat content, salt content, and mass remain physically consistent without artificial drifts. Sea ice concentration across individual thickness categories is jointly updated by the SST assimilation, allowing SST observations to influence the sea ice component at the DA step (Kimmritz et al., 2018). After assimilation, a post-processing step ensures the physical consistency of ocean state variables and updates other sea ice state variables. For instance, the volume of each sea ice category is proportionally scaled according to the updated sea ice concentration (Kimmritz et al., 2018, 2019). This implies that SST observations are used to update the ocean and sea ice components at every monthly assimilation step. The other components (e.g., atmosphere and land) are adjusted via the coupler during the model integration between the assimilation steps. Overall, this reanalysis system falls under the category of a semi-coupled DA system (Penny et al., 2017).

Observation errors are assumed to be uncorrelated in NorCPM. However, this assumption is not valid for the SST product assimilated (e.g., HadISST2.1, Section 2.2) because it inherently includes spatial correlations. To address this issue and minimize the impact of correlated observation errors, we decided to assimilate only the nearest SST data for each model grid cell. This assimilation strategy has been widely used in our previous studies (e.g., Counillon et al., 2016; Wang et al., 2019; Bethke et al., 2021).

The CoRea1860+ reanalysis is branched from a 30-member historical simulation in January 1860 (Section 2.1). To ensure a smooth start of the reanalysis from the historical ensemble simulation, the observation error variance is inflated by a factor of 8 during the first assimilation update. This inflation factor is gradually reduced by 1 after every two monthly assimilation updates until it reaches a value of 1 (Sakov et al., 2012; Counillon et al., 2016).

Assimilation systematically shrinks the ensemble spread, which may cause the ensemble to collapse. Several inflation techniques are employed to maintain the ensemble spread throughout the reanalysis production. The DEnKF inherently reduces the need for inflation to some extent. Additionally, we apply the moderation technique proposed by Sakov and Oke (2008): while the ensemble mean is updated using the original observation error variance, the ensemble spread is updated with an observation error variance inflated by a factor of 4. Furthermore, to prevent strong updates, we inflate the observation error variance to ensure that the analysis remains within two standard deviations of the background error. These measures help to sustain ensemble spread and enhance the reliability of the reanalysis system (Section 4.1).

## 3  Reference datasets and metrics

In this section, we present reference datasets and metrics used to evaluate the CoRea1860+ reanalysis. Since the CoRea1860+ reanalysis spans the period from 1860 to the present, we mostly use long-term independent datasets and observations to validate our reanalysis. We selected 11 datasets consisting of historical reconstructions for the atmosphere, ocean, and sea ice components, besides the RAPID observations (Table 1).

### 3.1  Reference datasets

**20CRv3** (Slivinski et al., 2021) is a global atmospheric reanalysis from 1806 to 2015 and consists of 80 ensemble members. It has been produced by assimilating only surface pressure observations and prescribing SST and sea ice concentration. 20CRv3 performs well not only on weather time scales but also on climate time scales. **ERA-20C** (Poli et al., 2016) is the first atmospheric reanalysis of the $20^{th}$ century of ECMWF. It spans from 1900 to 2010 with a single member. It assimilated observations of surface pressure and surface marine wind.

**ORA-20C** (de Boisséson et al., 2018) is an ocean reanalysis of ECMWF and covers the period 1900–2009. It used the atmospheric forcing from ERA-20C and assimilated temperature and salinity profile data. **SODA2.2.4** (Giese and Ray, 2011) is an ocean reanalysis available from 1871 to 2010. It was forced by a former version of 20CRv3 (i.e., 20CRv2) and assimilated both surface and subsurface ocean observations. **CHOR** and **CHORE** (Yang et al., 2017) are two historical ocean reanalyses from 1900 to 2010. Both reanalyses assimilated hydrographic profile data with the 3DVAR assimilation scheme and SST data via the nudging scheme. While CHOR was forced by a former version of 20CRv3, CHORE was forced by ERA-20C. **EN4.2.2** (Good et al., 2013) is an ocean objective analysis that relies on statistical interpolation of the quality-controlled hydrographic profile data. It provides gridded monthly average estimates of the ocean and covers the period from 1900 to the present. It is relaxed to the monthly climatology defined over 1971–2000 in the absence of any observations.

**Table 1.** Reference datasets used in the evaluation of the reanalysis CoRea1860+.

| Dataset | Component | Period | Type | Category | Producer | Reference |
|---------|-----------|--------|------|----------|----------|-----------|
| 20CRv3 | atmosphere/land | 1806–2015 | reanalysis | surface input | NOAA[*] | Slivinski et al. (2021) |
| ERA-20C | atmosphere/land | 1900–2010 | reanalysis | surface input | ECMWF[+] | Poli et al. (2016) |
| ORA-20C | ocean | 1900–2009 | reanalysis | subsurface input | ECMWF | de Boisséson et al. (2018) |
| SODA2.2.4 | ocean | 1871–2010 | reanalysis | surface and subsurface input | U. Maryland[†] | Giese and Ray (2011) |
| CHOR | ocean | 1900–2010 | reanalysis | surface and subsurface input | CMCC[‡] | Yang et al. (2017) |
| CHORE | ocean | 1900–2010 | reanalysis | surface and subsurface input | CMCC | Yang et al. (2017) |
| EN4.2.2 | ocean | 1900– | objective analysis | subsurface input | Met Office | Good et al. (2013) |
| RAPID | ocean | 2004–2021 | observation | subsurface input | NOC[◇] | Moat et al. (2024) |
| HadISST2.2 | sea ice | 1850–2019 | objective analysis | surface input | Met Office | Titchner and Rayner (2014) |
| IAPICE1 | sea ice | 1901–2019 | objective analysis | surface input | IAP[○] | Semenov et al. (2024) |
| SIBT1850 | sea ice | 1850–2017 | objective analysis | surface input | NOAA | Walsh et al. (2017) |
| CERA-20C | atmosphere/land/ ocean/sea ice | 1901–2010 | reanalysis | surface and subsurface input | ECMWF | Laloyaux et al. (2018) |

[*] National Oceanic and Atmospheric Administration [+] European Centre for Medium-Range Weather Forecasts [†] University of Maryland [‡] Euro-Mediterranean Center on Climate Change [◇] National Oceanography Centre [○] Institute of Atmospheric Physics at Chinese Academy of Sciences

**RAPID** (Moat et al., 2024) makes use of arrays of moorings to monitor the variability of the meridional overturning circulation at $26°$ N in the Atlantic and has sustained the observations since 2004. It has underpinned a new understanding of the large-scale ocean circulation in the North Atlantic.

**HadISST2.2** (Titchner and Rayner, 2014) contains monthly mean sea ice concentrations on a $1°$ grid from 1850 to 2019. It combined passive microwave data with historical sources, such as sea ice charts. In periods with insufficient observations, e.g., before 1900 and in the 1940s in the Arctic and before 1940 in the Antarctic, the HadISST2.2 data are replaced by climatology (i.e., constant). **IAPICE1** (Semenov et al., 2024) is a newly developed $1° \times 1°$ gridded dataset providing monthly Arctic sea ice concentration for the period 1901–2019. It is constructed by decomposing monthly surface air temperature over land, SST, and sea level pressure fields into empirical orthogonal functions. Multiple regression models are then employed to predict the principal components of sea ice concentration using the principal components of surface air temperature, SST, and sea level pressure as predictors. **SIBT1850** (Walsh et al., 2017) is a monthly gridded Arctic sea ice concentration product back to 1850 and synthesizes the historical observations from different sources: ship observations, compilations by naval oceanographers, analyses by national ice services, satellite passive microwave data, and others. Monthly sea ice concentration is given in a $1/4°$ horizontal resolution.

**CERA-20C** (Laloyaux et al., 2018) is a coupled reanalysis of the $20^{th}$ century produced by ECMWF and has 10 ensemble members. SST was relaxed toward the HadISST2.1 monthly ensemble product in the ocean component. Hydrographic profile data from the EN4.0.2 dataset, surface pressure observations from the International Surface Pressure Databank, and marine wind observations from ICOADS were assimilated within a fully-coupled DA framework. Compared to the uncoupled ocean ORA-20C and the atmospheric historical reanalysis ERA-20C, CERA-20C represents better atmosphere-ocean heat fluxes and

sea level pressure variations (Laloyaux et al., 2018). The period of 1900–2010 was divided into 14 different production streams
of 10 years. All streams were initialized from ERA-20C and ORA-20C reanalyses, and produced in parallel. The first two
years of each stream were considered as the spin-up period and discarded to generate CERA-20C over 1901–2010. CERA-20C
shows discontinuities between streams in the slow components (e.g., Figure 10 in Laloyaux et al., 2018). Therefore, only its
atmospheric data are used in this study to validate our reanalysis.

## 3.2   Metrics

We present and compare the time series from CoRea1860+ and reference datasets to give an overview evaluation of global or
regional climate variability. In terms of statistics on grid points, we interpolate data to a common and regular 5°x5° grid (except
2°x2° for sea ice concentration) and then use the anomaly correlation coefficient (ACC) to assess the reanalysis performance.
The ACC is the correlation between anomalies of our reanalysis and anomalies of the reference values and is defined as follows:

$$\text{ACC} = \frac{\sum (x - \overline{x})(y - \overline{y})}{\sqrt{\sum (x - \overline{x})^2} \sqrt{\sum (y - \overline{y})^2}}, \tag{1}$$

where $x$ are the anomalies of our reanalysis (ensemble mean) and $y$ are the anomalies of the reference values. $\overline{x}$ and $\overline{y}$ are the
mean values averaged over time. The sum $\sum$ is here computed over time. The period used to compute ACC corresponds to the
overlapping period between CoRea1860+ and the specific reference dataset and thus varies across different reference datasets.

For the significance test of ACC, we use the significance level of $\alpha = 0.1$ and follow the methodologies of Yeager et al.
(2018) and Bethke et al. (2021). A bootstrap technique is employed to generate a probability distribution function of ACC
accounting for uncertainties arising from both temporal sampling and the limited ensemble size. Specifically, we generate
1000 bootstrapped ACCs for each tested ACC by resampling the data using $x$-$y$ pairwise sampling with replacement in 5-year
blocks and resampling ensemble members (also with replacement). In cases where the fraction of bootstrapped ACCs with an
opposite sign to the tested ACC is larger than $\alpha$, the tested ACC is assumed to not differ significantly from zero. Grids that fail
the significance test are marked with a slash on the ACC maps.

In terms of the reliability of the reanalysis, we make use of the assimilation diagnostics that have been proposed by Desroziers
et al. (2005) and have been used in many applications (e.g., Bethke et al., 2018; Counillon et al., 2016; Sakov et al., 2012;
Slivinski et al., 2021). To do so, we define global statistics as follows:

$$\overline{d} = \sum wd, \tag{2}$$

$$\overline{\sigma_f} = \sqrt{\sum w\sigma_f^2}, \tag{3}$$

$$\overline{\sigma_o} = \sqrt{\sum w\sigma_o^2}, \tag{4}$$

$$\overline{\sigma_t} = \sqrt{\overline{\sigma_f}^2 + \overline{\sigma_o}^2}, \tag{5}$$

$$\hat{d} = \sqrt{\sum wd^2}, \tag{6}$$

$$\tag{7}$$

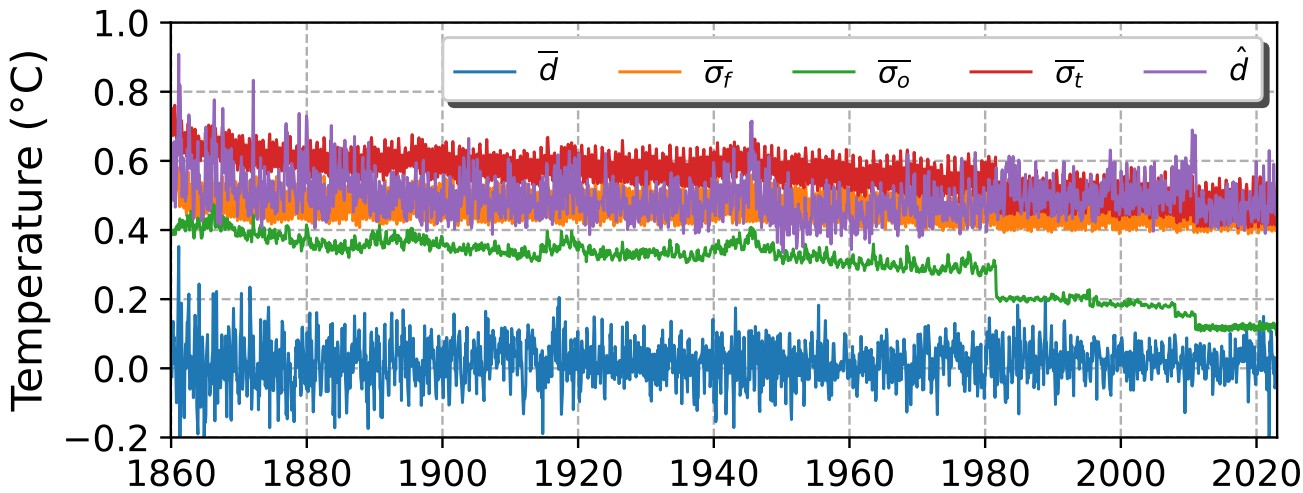

**Figure 1.** Global assimilation diagnostics for the assimilated variable – monthly SST anomalies: bias (blue line), background error (orange line), observation error (green line), total error (red line), and RMSE (purple line).

where $w$ is the area of the grid divided by the global ocean area, $d$ is the difference between the ensemble mean of the anomalies of our reanalysis and the reference anomalies, $\sigma_f$ is the standard deviation of the ensemble of the prior model state representing background error and $\sigma_o$ is the observation error. $\hat{d}$ represents the global RMSE computed in space. $\overline{\sigma_f}$ and $\overline{\sigma_o}$ represent the globally averaged background and observation errors, respectively. According to Desroziers et al. (2005), in the case where the observation and background errors are uncorrelated and unbiased and the system is reliable, the RMSE $\hat{d}$ is expected to be

equal to the total error $\overline{\sigma_t}$.

## 4   Evaluation

### 4.1   Reliability

According to the temporal evolution of $\overline{d}$ for SST (the assimilated variable, Figure 1), the system has larger $\overline{d}$ amplitudes about $[-0.2\,°\mathrm{C},\ 0.2\,°\mathrm{C}]$ over the period 1860–1880 and gradually converges to a stable interval $[-0.1\,°\mathrm{C},\ 0.1\,°\mathrm{C}]$. $\overline{d}$ varies around

zeros over the whole period and has no significant drift, indicating the reanalysis system is stable.

    The observation error (green line in Figure 1) slightly decreases until 1982, due to the increased sample size of observations and the improved data quality. It dramatically shrinks in 1982 thanks to the satellites that provide good global coverage of high-quality SST estimates. Another slight drop in the observation error occurs in 2011 due to the shift of the assimilated dataset from HadISST2.1 to OISSTV2 (Section 2). The background error (orange line in Figure 1) is quite stable before 1982

and slightly decreases in the satellite era due to smaller observation errors (i.e., more accurate observations).

The RMSE $\hat{d}$ (purple line in Figure 1) varies mostly within the interval $[0.4\,°C,\ 0.6\,°C]$ and has no significant trend over the whole period, meaning the reanalysis system is stable. Some significant changes are related to the evolution of observation networks, e.g., 1982, and 2011 (i.e., the time of switching SST datasets, Section 2.2). The RMSE is relatively higher in El Niño or La Niña years, e.g., 1982/1983, 1997/1998, and 2009/2010. The total error (red line in Figure 1) is about $0.6\,°C$ with a slight decreasing trend before 1982 and about $0.5\,°C$ after 1982. Its shrinkage in 1982 is mainly because of introducing the satellite observations.

In terms of reliability, the RMSE is equal to the total error if the system is perfectly reliable. In our system, $\overline{\sigma_t}$ is slightly higher than $\hat{d}$ before 1982 but the amplitudes of the two quantities match very well since 1982 when the global coverage and quality of observations are good. Several reasons can cause the mismatch between $\overline{\sigma_t}$ and $\hat{d}$ before the 1980s. As mentioned in Counillon et al. (2016), NorCPM has a slightly excessive ensemble spread, i.e., an underestimation of its accuracy. More-over, we solely update the ocean component of our system, keeping the atmosphere unchanged at the assimilation step. This inevitably results in assimilation shocks and increased variability. We use a deterministic variant of the EnKF (DEnKF) that inherently inflates the analysis error covariances (Sakov and Oke, 2008) and ad-hoc inflation methods proposed by Sakov et al. (2012), which maintain our system in an overdispersive regime. Sakov and Oke (2008) have found that the system is prefer-able to be overdispersive than underdispersive. Additionally, before the satellite era, the assimilated product – HadISST2.1 – relies on interpolating and extrapolating sparse in situ observations based on the physical constraints, making the observa-tion and background errors not fully uncorrelated. Overall, the reanalysis system is stable and reliable, in particular in the observation-rich period.

## 4.2   Ocean variability

Since SST in CoRea1860+ is constrained by DA and closely follows the assimilated SST dataset (Bethke et al., 2021), we have chosen not to include an evaluation of SST variability. This section presents the temporal evaluation of two other critical ocean variables: ocean heat content (OHC) and the Atlantic meridional overturning circulation (AMOC). The OHC serves as a key indicator of the ocean's role in heat uptake, storage, and redistribution, making its evolution particularly relevant to understanding climate variability (de Boisséson et al., 2018). The AMOC, on the other hand, is a large-scale circulation pattern in the Atlantic Ocean, playing a pivotal role in climate regulation due to its ability to transport heat, freshwater, and carbon across the globe (Carton and Hakkinen, 2011). By redistributing heat between the equator and higher latitudes, the AMOC has a profound impact on weather systems, regional climates, and long-term variability such as AMV (Zhang et al., 2019). However, its evolution during the historical era remains poorly understood, primarily due to the scarcity of direct current measurements, which have only been available consistently at $26°\,N$ since 2004 with the advent of the RAPID program.

### 4.2.1   Ocean heat content

We evaluate three ocean regions: the open-water ocean $[60°S, 60°N]$, the Arctic Ocean $[60°N, 90°N]$, and the Southern Ocean $[60°S, 90°S]$. In each region, we assess OHC in both the upper 0–300 m and 0–2000 m layers. The two polar regions are of particular interest, as they are critical for climate studies but were poorly observed before the satellite era. Moreover, SST data

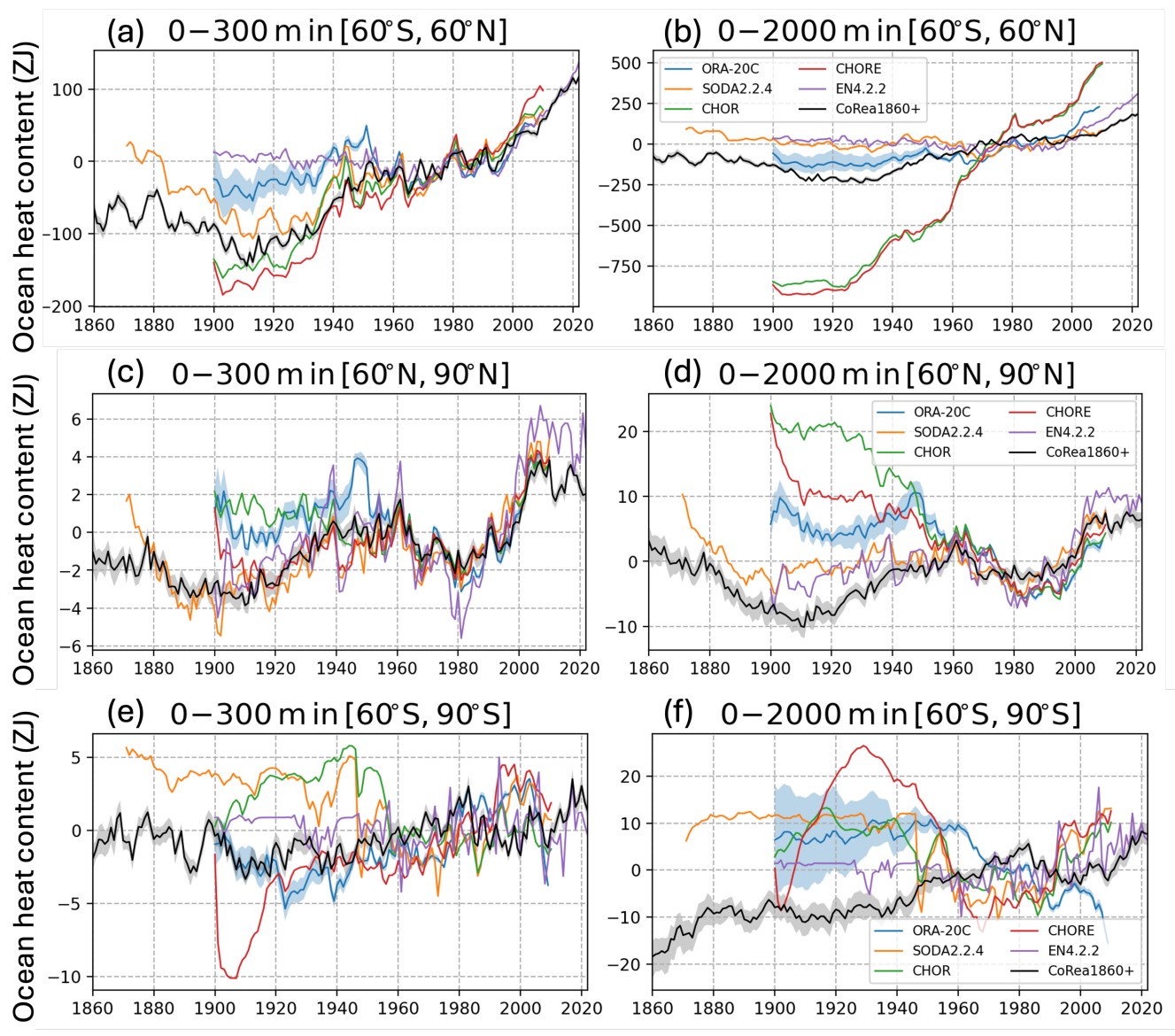

**Figure 2.** Time series of the anomalies of OHC in 0-300 m (left column) and 0-2000 m (right column) in [60°S, 60°N], [60°N, 90°N], and [60°S, 90°S]. The anomaly is relative to the climatology for 1950–2009. The unit is ZJ ($10^{21}$ J).

in areas covered by sea ice are not assimilated, making it especially valuable to evaluate the system's performance under sea
ice conditions.

For the OHC in the upper 0–300 m over the region [60°S, 60°N], the datasets reveal notable differences before 1940 (Figure 2a). However, all datasets—except EN4.2.2— suggest an OHC decline in the early 1900s and a warming trend over 1920–1940. EN4.2.2 uses relaxation towards the observed 1971-2000 climatology when no hydrographic profile data are available. The scarcity of profile observations during this period explains the very limited multidecadal variability between 1900 and 1950.
From the 1950s onward, all datasets align better with each other, reflecting a global warming hiatus from 1950 to 1980, followed by a strong warming trend beginning in 1990. Overall, CoRea1860+ exhibits variability that falls within the range of the other datasets. Before 1900, only SODA2.2.4 and CoRea1860+ are available and show some slight disagreement. SODA2.2.4 shows a comparable heat content to that of the 1980s while CoRea1860+ shows a lower level (albeit higher than in 1900-1920).

For the OHC in the upper 0–300 m over the region [60°N, 90°N] (Figure 2c), CoRea1860+ aligns with SODA2.2.4 starting
in 1880, following the spin-up period of SODA2.2.4, and with CHORE starting in 1910, after the spin-up of CHORE. It exhibits similar long-term variability to EN4.2.2 but with slightly weaker magnitudes. It should be highlighted that the hydrographic profile data availability at high latitudes is higher. Over the period 1980-2010, EN4.2.2 depicts a higher warming trend in the Arctic than in the open-water ocean ([60°S, 60°N]), which confirms the polar amplification phenomenon. Additionally, while CoRea1860+, SODA2.2.4, EN4.2.2, and CHORE display negative anomalies during 1900–1950, ORA-20C and CHOR show
positive anomalies during the same period. ORA-20C captures the early warming trend during this time, whereas CHOR does not. From the 1950s onward, all datasets are in good agreement in temporal variability, while EN4.2.2 shows higher variability amplitude than the other products.

For the OHC in the upper 0–300 m over the Southern Ocean ([60°S, 90°S], Figure 2e), the datasets show greater discrepancies compared to the other two regions, particularly before 1960. For instance, CHOR and SODA2.2.4, which were forced by a
375 former version of the atmospheric reanalysis 20CRv3, exhibit strong positive anomalies. In contrast, ORA-20C and CHORE, forced by the atmospheric reanalysis ERA-20C, display negative anomalies. EN4.2.2 remains close to climatology before 1960 due to the sparsity of profile observations. After 1960, the differences among the datasets diminish to some extent. Overall, CoRea1860+ exhibits a multidecadal variability with a slight warming trend over the entire period. Notably, CoRea1860+ appears to align more closely with EN4.2.2 after 1960.

For the OHC in the upper 0–2000 m over the region [60°S, 60°N], significant differences are observed across datasets (Figure 2b). CHOR and CHORE demonstrate similar OHC variability and exhibit a pronounced increasing trend, consistent with the findings of Yang et al. (2017) (their Figure 10c). In contrast, CoRea1860+ and ORA-20C show moderately increasing trends accompanied by clear multidecadal variability. Meanwhile, EN4.2.2 shows neither multidecadal variability nor an increasing trend before 2000. This is again caused by the relaxation to climatology when no observations are available. SODA2.2.4 shows
weak multidecadal variability, as it assimilates SST data to constrain only mixed layer properties, e.g., temperature and mixed layer depth (Giese and Ray, 2011), when no hydrographic profile data are available. Post-2000, all datasets exhibit a strong global warming trend. Notably, CoRea1860+ displays a weaker warming trend than EN4.2.2 but aligns with SODA2.2.4. The other datasets show a stronger warming trend than EN4.2.2.

For the OHC in the upper 0–2000 m over the region [60°N, 90°N] (Figure 2d), all datasets show consistent multidecadal variability and a slight warming trend from the 1950s onward. Before 1950, CoRea1860+ exhibits multidecadal variability similar to ORA-20C, SODA2.2.4, and EN4.2.2, but with slightly stronger negative anomalies compared to SODA2.2.4 and EN4.2.2. In contrast, ORA-20C, CHOR, and CHORE demonstrate significant positive anomalies during 1900–1950 and CHOR and CHORE exhibit a declining trend over this period.

For the OHC in the upper 0–2000 m over the Southern Ocean ([60°S, 90°S], Figure 2f), the datasets show great discrepancies. Before 1960, while CHOR, SODA2.2.4, ORA-20C, and CHORE exhibit strong positive anomalies, EN4.2.2 remains close to climatology due to the sparsity of profile observations. After 1960, the differences among the datasets diminish to some extent, but clear conclusions remain challenging. Over the entire period, CoRea1860+ exhibits a weak multidecadal variability with a prominent warming trend.

For the annual variability of OHC in the upper 300 m (Figure 3), the spatially averaged ACCs between CoRea186+ and the comparison datasets range from $0.27$ to $0.47$, with the highest mean ACC observed in comparison with ORA-20C. CoRea1860+ demonstrates high ACCs in most ocean regions, attributed to similar SST constraints. However, ACCs are low or even negative in specific regions such as the Arctic Ocean, the Southern Ocean, and the eastern tropical Atlantic. The eastern tropical Atlantic stands out as a region where CoRea1860+ exhibits significant discrepancies compared to the other ocean reanalyses. This is primarily due to the model's difficulty in simulating eastern boundary upwelling systems (Richter, 2015). Richter (2015) identified notable biases in CMIP5 models, attributing them to several factors, including the underestimation of stratocumulus cloud cover, weaker-than-observed wind stress, unresolved offshore transport by mesoscale ocean eddies, and an overly diffuse vertical temperature gradient separating the warm upper ocean layer from the deeper ocean. There is a large disagreement in sea ice-covered areas, which is expected as there are nearly no observations there. The spatial patterns of ACCs with CHORE and CHOR are very similar, except in the Southern Ocean, likely because both datasets use the same model and observations but rely on different atmospheric forcings. ACCs with EN4.2.2 exhibit smoother spatial patterns and smaller magnitudes, primarily due to the absence of the model and the sparse availability of hydrographic profile data before 1950 (Figure 2a). In contrast, prominent positive ACCs with ORA-20C are found in open-water regions, including the North Atlantic extending into the Norwegian Sea. However, there is a strong negative correlation in the northern Indian Ocean that is not seen with the other datasets. ACCs with SODA2.2.4 are mostly positive in the open-water regions.

Unlike the reference datasets, CoRea1860+ does not utilize hydrographic profile data. However, it demonstrates significant positive ACCs with these datasets for OHC in the upper 2000 m across most ocean regions (Figure 3). The spatially averaged ACCs range from $0.19$ to $0.35$. The spatial patterns of ACCs with CHORE and CHOR are highly similar, with high ACCs observed in most open-water regions—except in the tropical Atlantic—and low ACCs in the Arctic Ocean and the Southern Ocean. ACCs with EN4.2.2 display smoother spatial patterns with smaller magnitudes. They are notably higher in the tropics and the Atlantic-Arctic region. Significant positive ACCs with ORA-20C are primarily concentrated in the tropics (except the tropical Atlantic) and mid-latitudes. Similarly, ACCs with SODA2.2.4 are high in the ENSO region, the Indian Ocean, and the Subpolar Gyre in the North Atlantic, while low ACCs are found in the South Atlantic and the Southern Ocean.

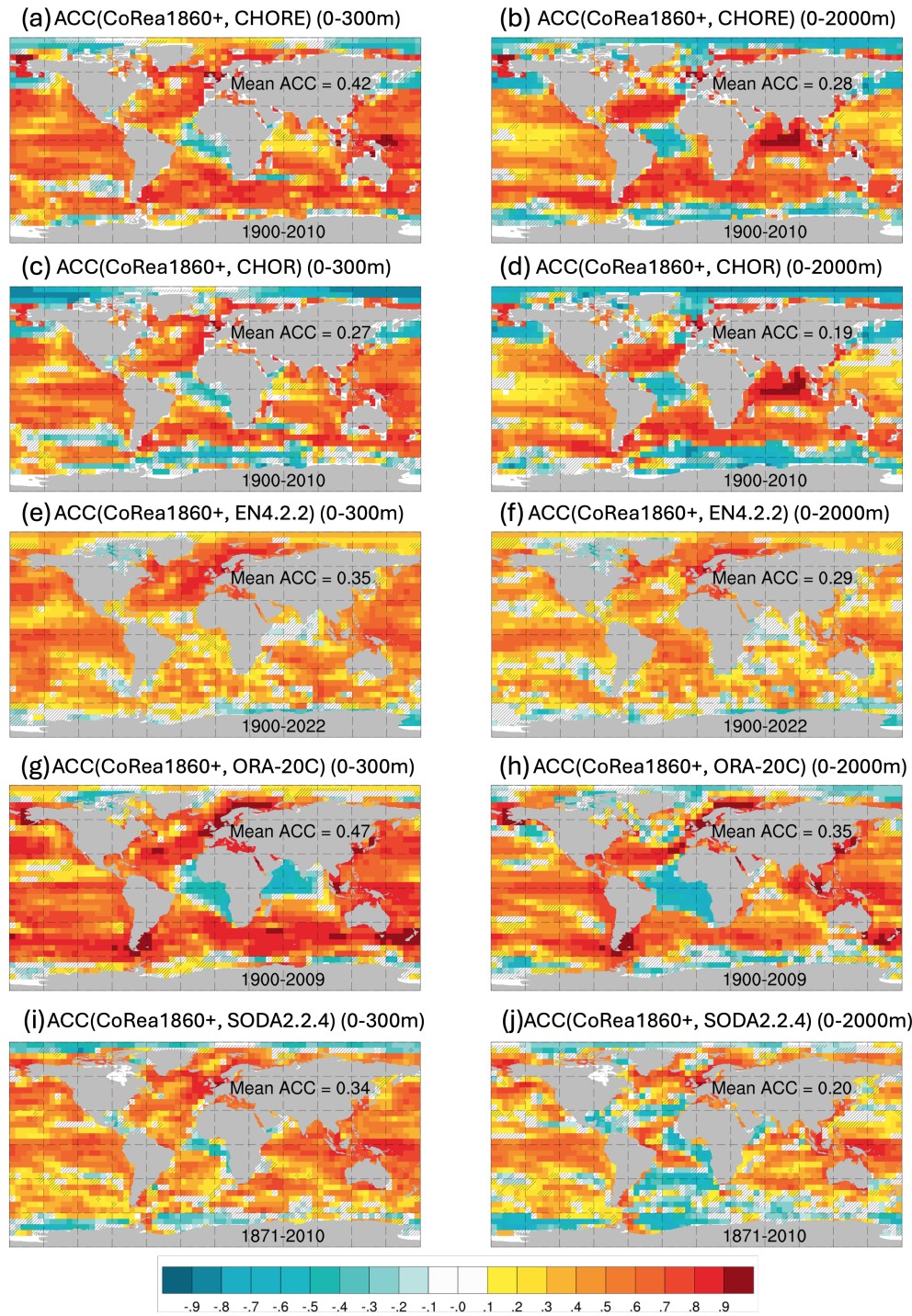

**Figure 3.** ACC of yearly OHC in 0-300m or 0-2000m of CoRea1860+ against different reference datasets. Grids that fail the significance test are marked with a slash. The period used to compute ACC corresponds to the overlapping period between CoRea1860+ and the specific reference dataset and is shown in Antarctica. The spatially averaged ACC is shown in Eurasia.

**Table 2.** Temporal mean and standard deviation of the maximum AMOC transport at 26° N over 2005–2021 for RAPID and 1950–2009 for the other datasets.

| Dataset | Mean (Sv) | Standard deviation (Sv) |
|---|---|---|
| ORA-20C | 11.4 | 1.7 |
| SODA2.2.4 | 16.2 | 2.5 |
| CHOR | 15.6 | 2.0 |
| CHORE | 16.0 | 2.0 |
| RAPID | 16.7 | 1.4 |
| CoRea1860+ | 32.0 | 1.8 |

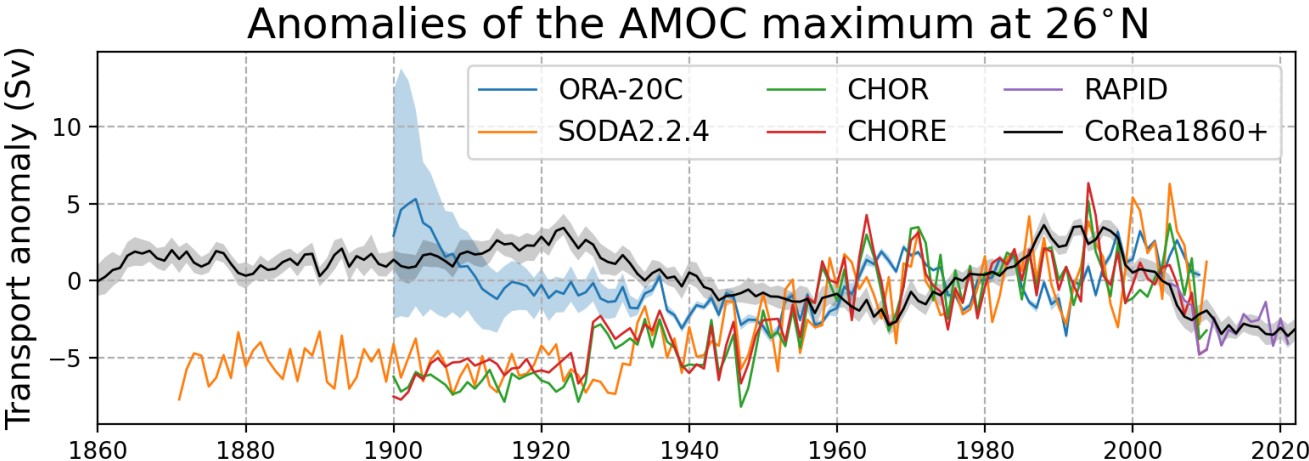

**Figure 4.** Time series of the anomalies of the maximum AMOC transport at 26° N. The anomaly is relative to the climatology for 1950–2009. RAPID is adjusted to the same level as CoRea1860+.

Overall, when compared to the other datasets, CoRea1860+ captures key multidecadal variability and long-term warming trends. In most cases, it shows significant positive ACCs with existing datasets in the representation of OHC variability, despite negative or non-significant ACCs arising from the lack of assimilation of observational data (e.g., SST data under sea ice and subsurface data). Note that interannual variability in OHC contributes much more to the computation of ACCs than the long-term trend (not shown).

### 4.2.2 Atlantic meridional overturning circulation

The AMOC is defined as the zonally and vertically integrated meridional volume transport across a latitude section in the Atlantic Ocean. It is measured in Sverdrups ($10^6 \, \mathrm{m^3 \, s^{-1}}$), and its magnitude varies with both latitude and depth. Due to the limited availability of long-term continuous measurements, the AMOC variability has only been consistently observed at 26° N

since 2004 (i.e., RAPID, Moat et al., 2024) and in a section of the Subpolar North Atlantic since 2014 (i.e., OSNAP, Fu et al., 2023). In this study, we focus on the RAPID measurement since the time series of OSNAP is too short. While CoRea1860+ exhibits a standard deviation in maximum AMOC transport at 26° N within the range of that of the reference datasets over 1950–2009 (except for RAPID over 2005–2021), its mean AMOC strength at 26° N is notably higher, with a time-mean of 32.0 Sv (Bentsen et al., 2013), exceeding that of the other datasets (Table 2). In the following, we focus on time series of the anomalies of the maximum AMOC transport at 26° N (Figure 4) and discuss the long-term trend and multidecadal variability of the AMOC. Note that the contribution of the Ekman component to the ensemble mean AMOC in CoRea1860+ is expected to be minimal due to the lack of wind synchronization in the assimilation system. We do not remove the Ekman component from the other reanalyses and RAPID; however, a fairer comparison of interannual variability across datasets would require doing so.

The AMOC in CoRea1860+ shows a slight declining trend over the last 160 years (Figure 4), consistent with findings from the other CMIP5 models, which generally exhibit a weak decline throughout the $20^{th}$ century (Cheng et al., 2013), as well as with AMOC reconstructions based on proxy data (Rahmstorf et al., 2015; Caesar et al., 2018). SODA2.2.4, CHOR, and CHORE reanalyses show a relatively strong increasing trend, which may relate to the choice of the initialization strategy at the start of the reanalysis and the fact that they are produced by the uncoupled models. NorESM is designed to simulate the response to external forcings and NorCPM is initialized from its stable preindustrial control run.

The AMOC in CoRea1860+ remains relatively neutral before 1920 but begins to display pronounced multidecadal variability thereafter, characterized by a 60-year periodicity and an amplitude of approximately 3 Sv (Figure 4). The AMOC exhibits stronger phases in the 1920s and 1990s, while weaker phases are observed in the 1970s and 2020s. These variations are consistent with observed AMV that shows positive phases from around 1930-1960s and mid-1990s onwards, and negative phases from around 1900-1920s and 1960-1980s, supporting a leading role for AMOC in driving AMV (Gulev et al., 2013; Keenlyside et al., 2015; Zhang et al., 2019). The CoRea1860+ AMOC variations show limited agreement with common proxy reconstructions of AMOC based on oceanic and atmospheric parameters – the peaks in these reconstructions after the 1950s tend to lead those in CoRea1860+ by roughly a decade (Figure 1 of Sun et al., 2021). While the other reanalysis datasets show distinct peaks in the 1910s, 1970s, and 2000s, with weaker phases occurring in the 1950s and 1990s, the multidecadal variability of AMOC is much weaker. Disagreement among reanalysis products is consistent with previous findings (Karspeck et al., 2017; Jackson et al., 2019). This may relate to the fact that coupled ocean-atmosphere processes are essential in representing the multidecadal processes (Zhang et al., 2019) while the other products are forced by atmospheric reanalysis.

All datasets including CoRea1860+ consistently capture the rapid decline in the AMOC from 2005 to 2010, which is in agreement with the RAPID array observations. Importantly, CoRea1860+, despite not assimilating ocean subsurface data, demonstrates AMOC variability comparable to RAPID since 2005. This consistency underscores the robustness of CoRea1860+ and lends confidence to its AMOC reconstruction before the availability of extensive subsurface observations.

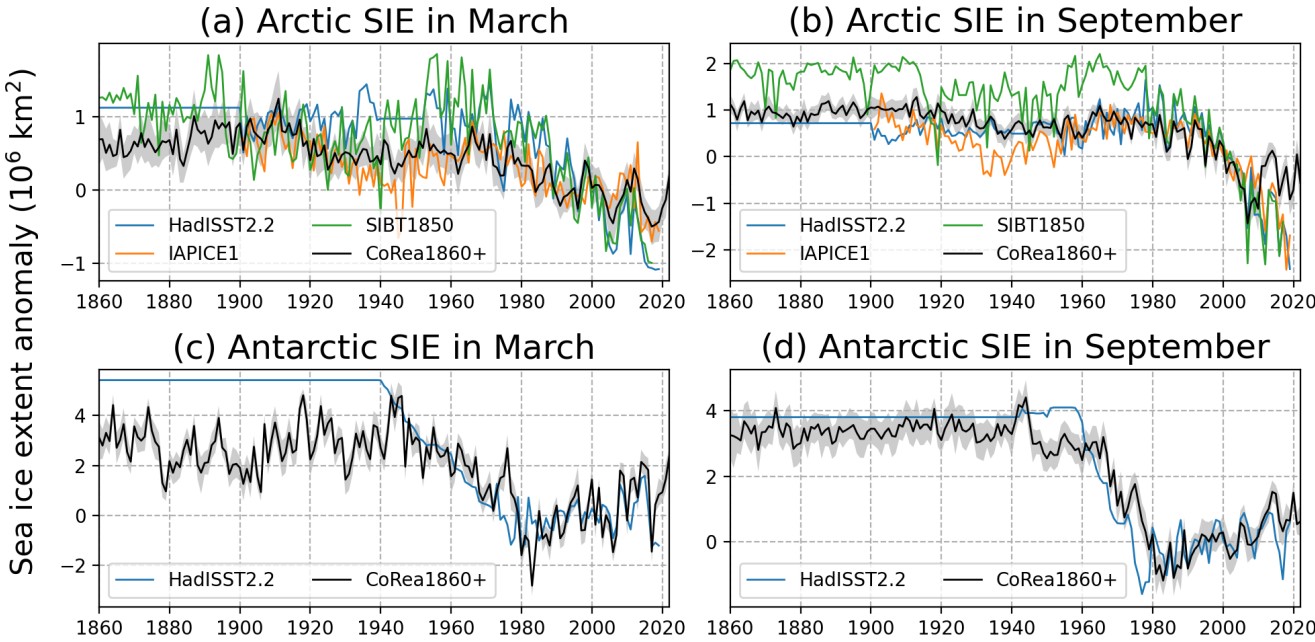

**Figure 5.** Time series of the anomalies of SIE in the Arctic (top) and Antarctic (bottom) in March (left) and September (right). The anomaly refers to the climatology from 1980 to 2017. Note that IAPICE1 and SIBT1850 provide sea ice data only in the Northern Hemisphere.

### 4.3 Sea ice variability

Sea ice variability is a key indicator of climate variability and change in the polar regions. It is highly related to the variability of OHC and AMOC (Delworth et al., 2016; Liu et al., 2020; Liu and Fedorov, 2022; Omrani et al., 2022; Oldenburg et al., 2024). This section presents the temporal evaluation of sea ice extent (SIE) and sea ice concentration in the Arctic and Antarctic. The SIE is defined as the area of the ocean where sea ice concentration is larger than 15 percent.

#### 4.3.1 Arctic sea ice variability

The Arctic SIE in March simulated by CoRea1860+, exhibits a long-term declining trend throughout the historical period, primarily attributed to anthropogenic forcings (Figure 5a). Superimposed on this decline, there is notable multidecadal variability, which reflects the internal variability of the climate system (Delworth et al., 2016; Omrani et al., 2022). Two periods of significant decline stand out: one in the early $20^{th}$ century and another in recent decades. IAPICE1 exhibits variability closely matching that of CoRea1860+ throughout the entire period, except for a slightly more pronounced decline in the early $20^{th}$ century. HadISST2.2 also displays a declining trend, particularly pronounced after 1980. However, HadISST2.2 is characterized by strong interannual variability but does not show comparable multi-decadal variability and the decline in the early $20^{th}$ century observed in the CoRea1860+ reanalysis. On the other hand, SIBT1850 shows both a long-term declining trend and

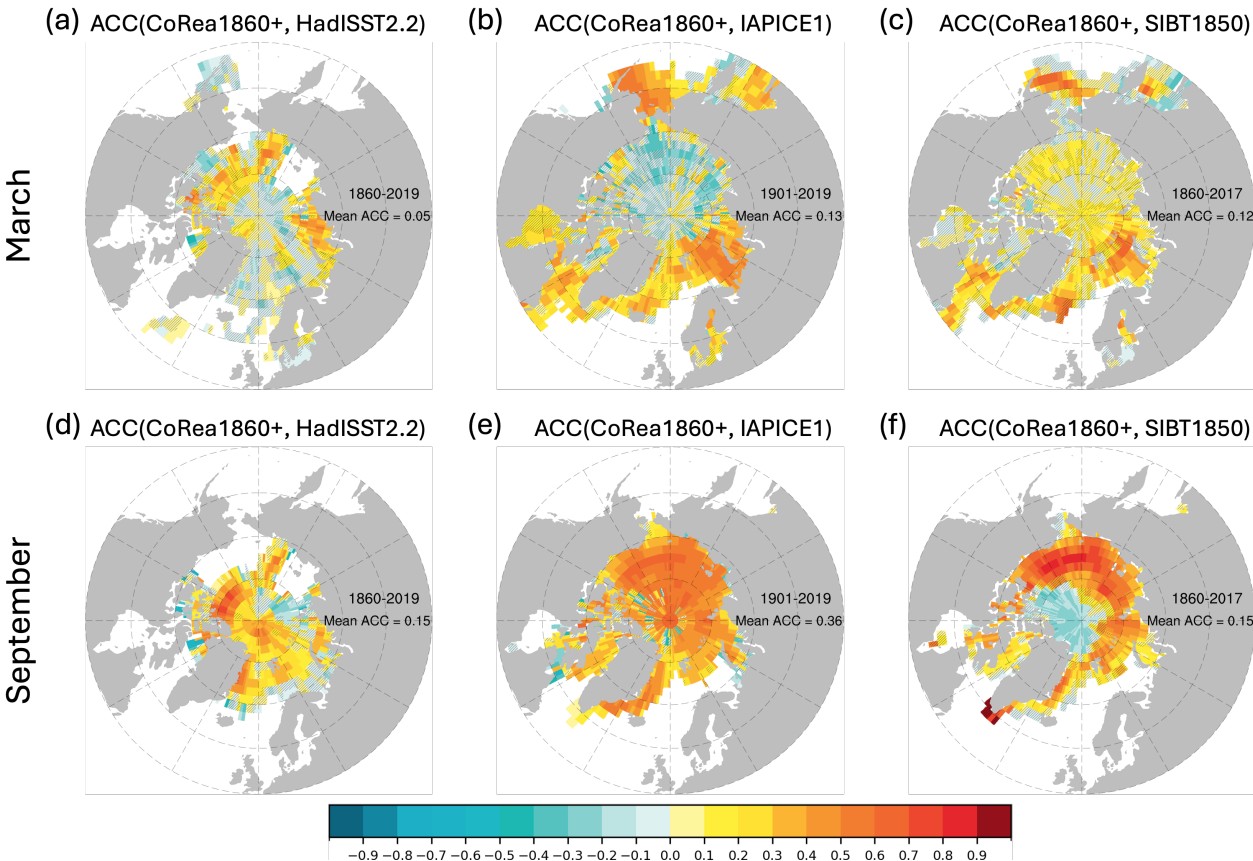

**Figure 6.** ACC of Arctic sea ice concentration in March or September of CoRea1860+ against HadISST2.2, IAPICE1, and SIBT1850. Grids that fail the significance test are marked with a slash. The period used to compute ACC corresponds to the overlapping period between CoRea1860+ and the specific reference dataset and is shown in Eurasia. The spatially averaged ACC is also shown in Eurasia.

significant multidecadal variability with a larger amplitude than CoRea1860+ and IAPICE1. In particular, CoRea1860+ underestimates the large SIE seen in SIBT1850 and HadISST2.2 in the 1960s and 1970s. In the satellite era, CoRea1860+ is close to IAPICE1 but underestimates the rate of decline in the Arctic SIE compared to both HadISST2.2 and SIBT1850. However, CoRea1860+ reproduces interannual variability similar to that observed in the other datasets, suggesting it reasonably captures short-term fluctuations in sea ice.

In terms of the ACCs of sea ice concentration in March (Figures 6a-c), CoRea1860+ demonstrates some agreement with the comparison datasets. The spatially averaged ACCs range from 0.05 to 0.13, with the highest mean ACC observed in comparison with IAPICE1. CoRea1860+ aligns with SIBT1850 and IAPICE1 in marginal ice regions, such as the Bering, Labrador, Greenland, and Barents Seas, while it shows agreement with HadISST2.2 in the Kara, East Siberian, and Beaufort Seas. Notably, the spatial coverage of ACC with HadISST2.2 is significantly lower compared to that with SIBT1850 and IAPICE1, maybe due to using climatological data before 1900 and in the 1940s when missing data (Figure 5).

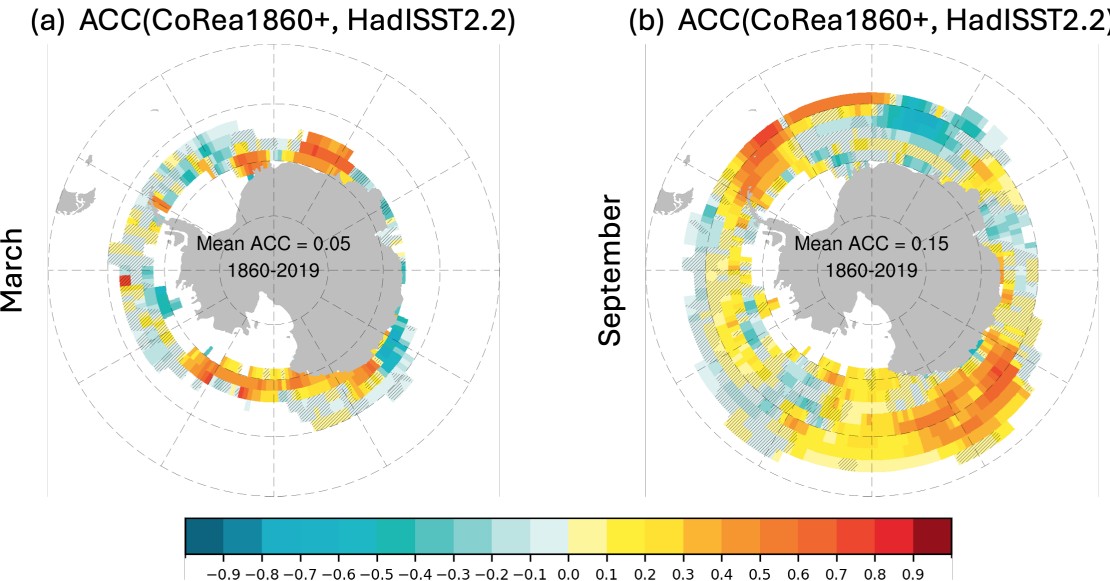

**Figure 7.** ACC of Antarctic sea ice concentration in March or September of CoRea1860+ against HadISST2.2. Grids that fail the significance test are marked with a slash. The period used to compute ACC corresponds to the overlapping period (i.e., 1860-2019) and is shown in Antarctica. The spatially averaged ACC is also shown in Antarctica.

The September Arctic SIEs in CoRea1860+ and HadISST2.2 exhibit similar behavior: moderate multidecadal variability
and a lack of any significant trend before 1980, followed by a strong declining trend in the satellite era (Figure 5b). The offset observed in the 2010s is primarily attributed to the switch in the assimilated dataset from HadISST2.1 to OISSTV2 (Section 2.2). IAPICE1 and SIBT1850 demonstrate the same variability as HadISST2.2 during the satellite era. While they show similar multidecadal variability before 1980, SIBT1850 exhibits stronger positive anomalies and larger variability compared to IAPICE1 (Semenov et al., 2024).

For the variability of sea ice concentration in September (Figures 6d-f), CoRea1860+ exhibits good agreement with IAPICE1 in almost the whole Arctic Ocean with the mean ACC of 0.36. It shows stronger agreement with SIBT1850 across most of the Arctic Ocean, except in the Central Arctic where ACCs are negative. Conversely, CoRea1860+ aligns with HadISST2.2 in the Central Arctic but exhibits weaker agreement in the marginal ice regions.

Note that interannual variability in sea ice concentration contributes much more to the computation of ACCs than the long-
500 term trend (not shown).

### 4.3.2 Antarctic sea ice variability

The Antarctic SIE in March (during the Austral summer) in CoRea1860+ exhibits more pronounced interannual-decadal variability compared to HadISST2.2, particularly before 1980 (Figure 5c). In contrast, HadISST2.2 shows constant but larger positive anomalies than CoRea1860+ before 1940. After 1940, the two datasets show a similar pattern: a pronounced decline

in SIE from 1940 to 1980, followed by a slight increase in the subsequent decades. In terms of the ACCs of sea ice concentration in March (Figure 7a), CoRea1860+ agrees well with HadISST2.2 in the Kong Hakon and Rose Seas. In the other regions, ACCs are either not significant or significantly negative. The spatially averaged ACC is about $0.05$.

The September Antarctic SIE in HadISST2.2 remains nearly constant until 1960, whereas CoRea1860+ exhibits notable interannual variability during the same period (Figure 5d). Unlike the differences observed in March, the anomalies of the two datasets before 1960 are relatively close in September. While HadISST2.2 begins a pronounced decline in SIE starting in 1960, CoRea1860+ shows a weaker decline from 1940 to 1980. After 1980, the September SIE variability in all datasets aligns closely, indicating better agreement in the recent decades. For the variability of sea ice concentration in September (Figure 7b), the spatially averaged ACC is about $0.15$. CoRea1860+ shows strong agreement with HadISST2.2 in the Atlantic and Pacific sectors and strong disagreement in the Kong Hakon Sea. In the other regions, ACCs are not significant.

The decline in Antarctic SIE from 1940 to 1980 is observed in both datasets during both summer and winter seasons, consistent with previous studies (e.g., Fogt et al., 2022; Dalaiden et al., 2023; Goosse et al., 2024; Divine et al., 2024). Fogt et al. (2022) have demonstrated statistically significant decreases in seasonal SIE during the early and middle $20^{th}$ century. Their reconstructions have relied on a principal component regression model that incorporated observations of land surface pressure and temperature across the Southern Hemisphere extratropics and midlatitudes (1905–2020), as well as indices of major climate modes, such as the Interdecadal Pacific Oscillation, the Southern Oscillation Index, and the Pacific Decadal Oscillation. Dalaiden et al. (2023) have used paleoclimate records from ice cores, tree rings, and DA to reconstruct long-term sea ice variability since 1700. They have identified a declining trend in the Weddell Sea throughout the $20^{th}$ century, with the largest decrease occurring before the 1960s. Goosse et al. (2024) have combined atmospheric pressure and temperature observations from 27 ground-based weather stations with outputs from nine large ensembles of coupled climate model simulations using offline DA. Their analysis has revealed a significant drop in Antarctic SIE at the end of the 1970s. Divine et al. (2024) have presented a dataset of marine climate, sea ice, and icebergs derived from logbooks of Norwegian whaling factory ships operating in the Southern Ocean between 1929 and 1940. This dataset includes approximately 4000 sea ice/open sea records from the Austral summer, suggesting a potentially higher seasonal SIE during the early 1930s. While these findings have provided important context, further investigation is needed to fully understand the strong decline in the mid-$20^{th}$ century, which lies beyond the scope of this study.

## 4.4 Atmosphere variability

As CoRea1860+ is a fully coupled reanalysis, we expect atmosphere variability to be improved by assimilating SST observations into the ocean and sea ice components via air-sea and air-sea ice interactions. This section presents the variability of key atmospheric variables in boreal winter (DJF) and summer (JJA), consisting of surface air temperature (SAT, i.e., air temperature at 2 m), total precipitation (PRECP), sea level pressure (SLP), and 500 hPa geopotential height (Z500).

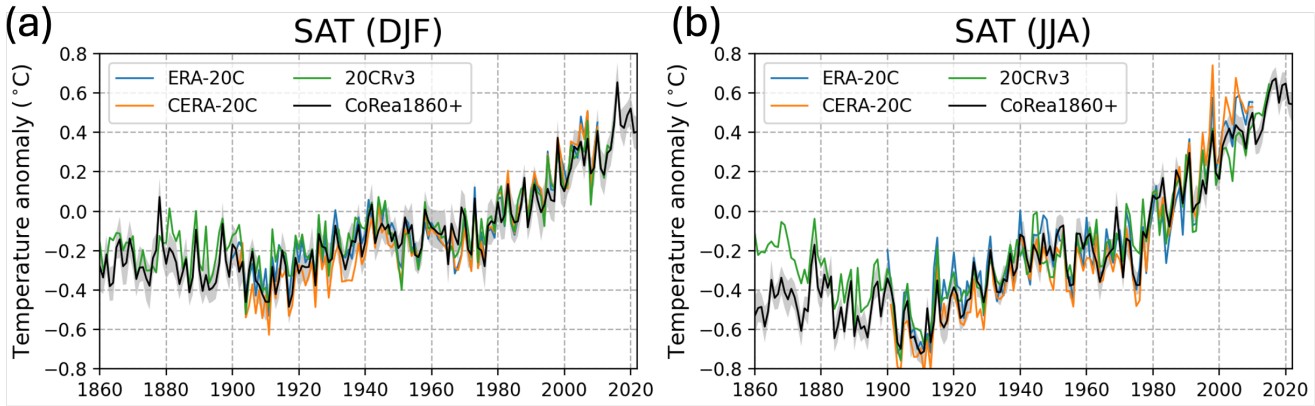

**Figure 8.** (a) Time series of the anomalies of SAT in DJF. (b) Time series of the anomalies of SAT in JJA. The anomaly is relative to the climatology for 1950–2009.

### 4.4.1 Surface air temperature

In terms of global mean SATs, CoRea1860+ aligns well with ERA-20C, CERA-20C, and 20CRv3 during overlapping periods, particularly in boreal winter (Figure 8a). In boreal summer, while there is a larger global warming trend and some differences between the datasets before 1900 and after 2000, they still exhibit very similar year-to-year variability (Figure 8b).

The good agreement in global mean SAT variability among the datasets is largely due to their reliance on similar SST datasets. CoRea1860+ assimilates historical SST data from HadISST2.1, while CERA-20C relaxes toward HadISST2.1. ERA-20C is forced by HadISST2.1, and 20CRv3 is mostly forced by SODAsi.3 (Giese et al., 2016). This connection is even more evident in the spatial distribution of ACCs (Figure 9).

CoRea1860+ exhibits consistency with ERA-20C, CERA-20C, and 20CRv3 in ocean regions and some land regions (Figure 9). The spatial patterns of ACCs are highly consistent across the reference datasets, with spatially averaged values ranging from 0.50 to 0.66. The highest mean ACC is obtained with CERA-20C, which is also a coupled reanalysis. In both boreal winter (DJF) and summer (JJA), ACCs are generally higher over oceans than over land where the oceanic influence is reduced. Due to stronger air-sea interactions in the tropics, ACCs are higher in lower latitudes compared to high latitudes, in particular over oceans. For instance, SST directly influences SLP, which drives changes in surface winds, and precipitation (Back and Bretherton, 2009; Gill, 1980; Lindzen and Nigam, 1987). In DJF, regions with lower or negative ACCs include Central Eurasia, Central Europe, western North America, southeastern South America, and areas near the Ross Sea in Antarctica. In JJA, lower ACCs are observed in North America, likely related to weaker large-scale atmospheric teleconnections from the tropics than in DJF (Wallace and Gutzler, 1981), and North Asia. Notably, ACCs are generally higher with CERA-20C and ERA-20C compared to 20CRv3. This is likely because CoRea1860+, CERA-20C, and ERA-20C all use the same SST dataset (i.e., HadISST2.1), whereas 20CRv3 uses mostly SODAsi.3 (Giese et al., 2016).

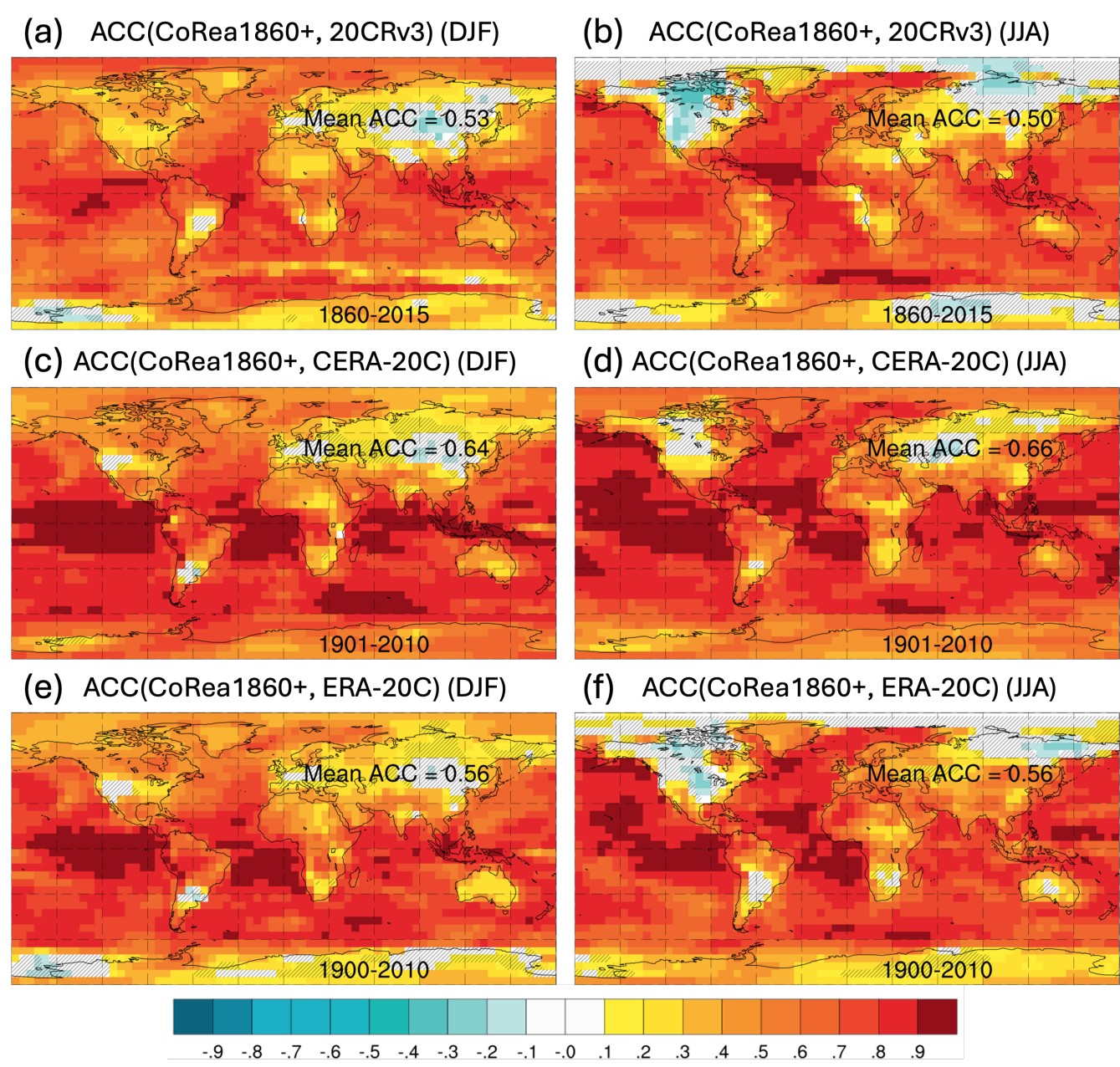

**Figure 9.** ACC of SAT in DJF or JJA of CoRea1860+ against 20CRv3, CERA-20C and ERA-20C. Grids that fail the significance test are marked with a slash. The period used to compute ACC corresponds to the overlapping period between CoRea1860+ and the specific reference dataset and is shown in Antarctica. The spatially averaged ACC is shown in Eurasia.

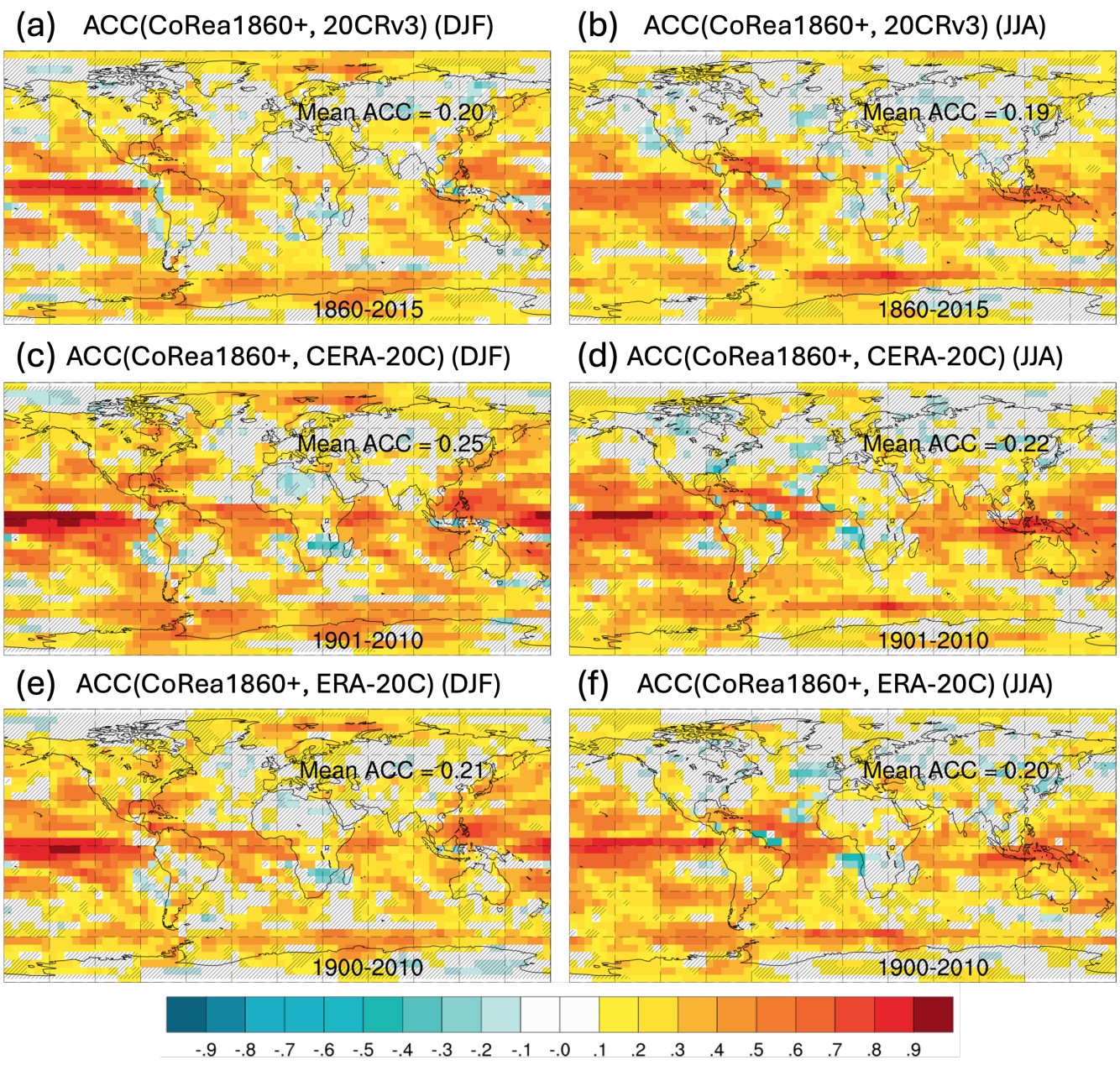

**Figure 10.** ACC of PRECP in DJF or JJA of CoRea1860+ against 20CRv3, CERA-20C and ERA-20C. Grids that fail the significance test are marked with a slash. The period used to compute ACC corresponds to the overlapping period between CoRea1860+ and the specific reference dataset and is shown in Antarctica. The spatially averaged ACC is shown in Eurasia.

### 4.4.2 Precipitation

For PRECP, the spatial patterns of ACCs are broadly similar in boreal winter (DJF) and summer (JJA), with some notable differences over land, such as in North America (Figure 10). ACCs are particularly high in the tropics because of strong air-sea interactions there, especially over the tropical Pacific due to the strong influence of ENSO. The agreement is higher between CERA-20C and CoRea1860+ (0.25 in DJF and 0.22 in JJA), which may relate to coupled processes. In the mid-latitudes, ACCs are generally higher in the Southern Hemisphere compared to the Northern Hemisphere, largely due to the greater ocean coverage in the south. Additionally, higher ACCs are observed near western boundary currents (Larson et al., 2024), such as the Gulf Stream extension, the Kuroshio–Oyashio Current, the Agulhas Current, and the Brazil–Malvinas Current where air-sea interactions modulate atmospheric variability (Minobe et al., 2008; Brayshaw et al., 2011; Hand et al., 2014; Omrani et al., 2019). ACCs are generally lower over land far away from the ocean, except for regions influenced by tropical Pacific teleconnections, such as North America in DJF (Wallace and Gutzler, 1981; Leathers et al., 1991) and West Africa in JJA (Janicot et al., 1996). In the Arctic, ACCs in DJF are higher than in JJA, particularly in the Norwegian-Barents Seas, primarily due to the relatively stronger influence of the Norwegian Current in winter (e.g., Wang et al., 2019). Despite differences in reference datasets, ACCs in the Southern Ocean remain high during both DJF and JJA, primarily due to the Antarctic Circumpolar Current.

### 4.4.3 Sea level pressure

For SLP in both boreal winter (DJF) and summer (JJA) (Figure 11), ACCs are particularly high in the tropics, especially over the tropical Pacific, due to the strong influence of ENSO and its teleconnections (e.g., Luo et al., 2005). In contrast, ACCs are lower in the mid-latitudes compared to the tropics, especially in regions such as North America, the North Atlantic, and North Asia. ACCs are low in the Arctic during both DJF and JJA. However, in the Antarctic, ACCs are relatively high during DJF but notably low during JJA. The spatially averaged ACCs are between 0.30 and 0.35 in DJF and between 0.20 and 0.27 in JJA.

It is worth noting that ERA-20C, CERA-20C, and 20CRv3 all assimilate surface pressure observations, whereas the CoRea1860+ reanalysis only assimilates SST observations. The influence of SST assimilation on SLP relies primarily on thermodynamic processes and air-sea interactions, which are dynamically consistent.

The North Atlantic Oscillation (NAO) is a predominantly atmospheric mode of climate variability and represents one of the most prominent patterns of climate fluctuations over the North Atlantic and surrounding regions (e.g., Hurrell, 1995). Following the definition in Hurrell (1995), we calculate the winter (December–March) NAO index as the difference between normalized SLPs at Lisbon and Stykkisholmur (Figure 12). The NAO indices from ERA-20C, CERA-20C, and 20CRv3 show strong agreement, reflecting the constraint provided by the assimilation of surface pressure observations (Figure 12a). In contrast, CoRea1860+, which assimilates only SST observations, exhibits weak interannual variability in the ensemble mean and a relatively large ensemble spread. The correlations between the NAO indices from CoRea1860+ and those from the reference reanalyses are all below 0.16. This indicates that SST assimilation alone is not sufficient to synchronize the NAO's year-to-year fluctuations with those in the other reanalyses. Nevertheless, CoRea1860+ shows the NAO's decadal variability (Figure

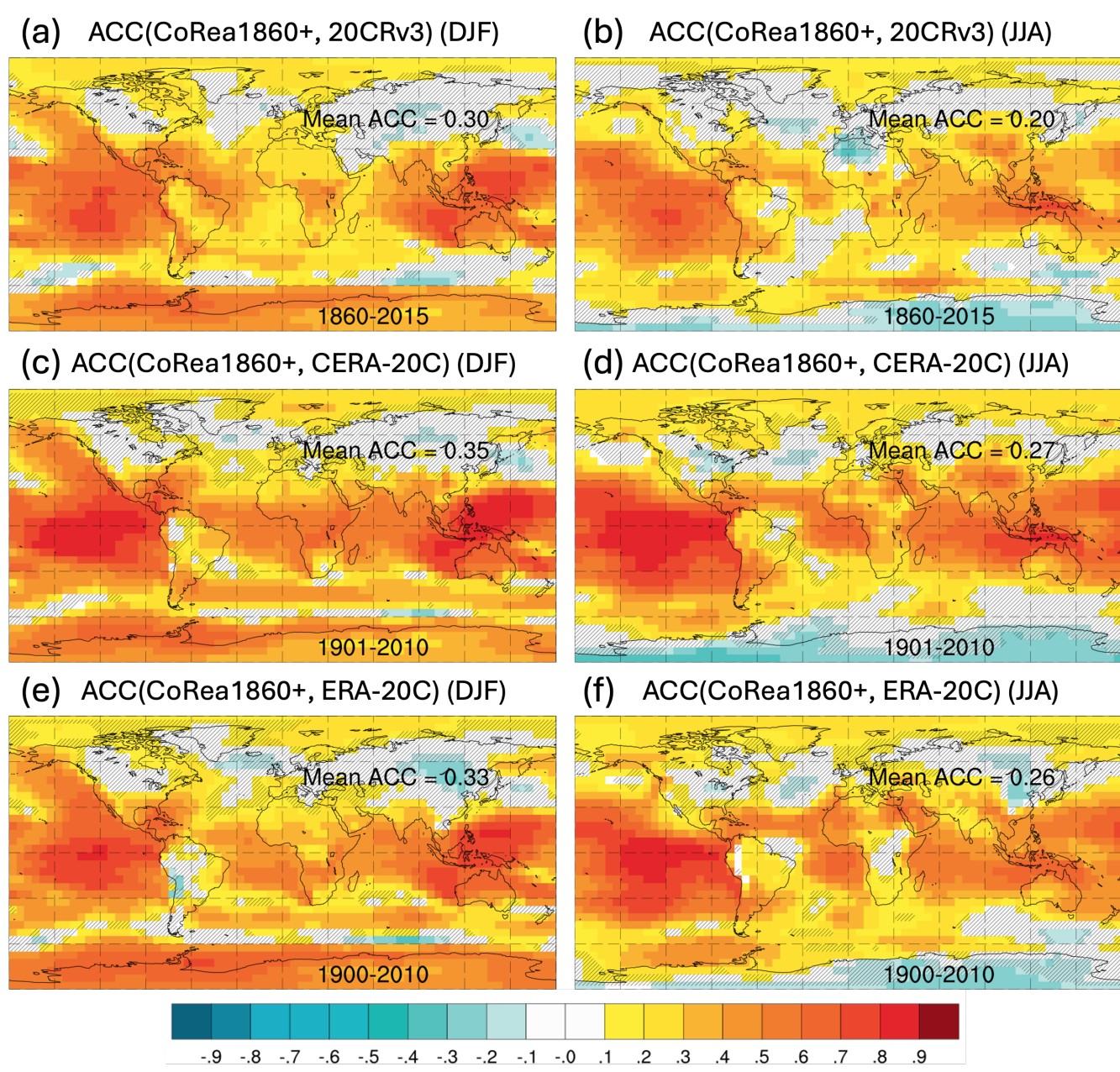

**Figure 11.** ACC of SLP in DJF or JJA of CoRea1860+ against 20CRv3, CERA-20C and ERA-20C. Grids that fail the significance test are marked with a slash. The period used to compute ACC corresponds to the overlapping period between CoRea1860+ and the specific reference dataset and is shown in Antarctica. The spatially averaged ACC is shown in Eurasia.

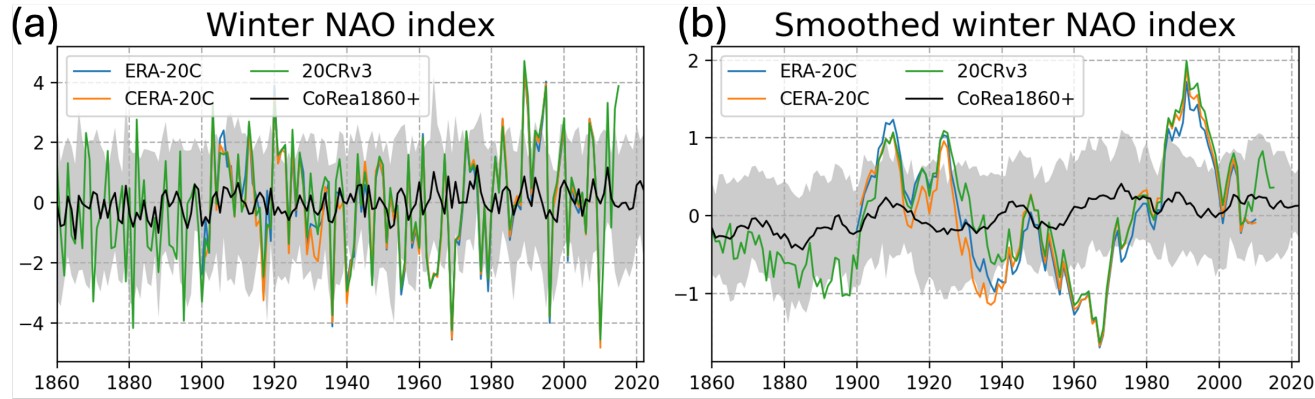

**Figure 12.** (a) Winter (December-March) station-based index of the NAO. (b) 10-year running average of the winter NAO index.

12b), suggesting a potential oceanic influence on longer timescales. This highlights an area of interest for future investigation, particularly in exploring how ocean-atmosphere coupling may contribute to low-frequency NAO variability.

### 4.4.4 500 hPa geopotential height

For Z500, the spatially averaged ACCs range from $0.36$ to $0.46$, with no substantial differences between DJF and JJA. In the tropics, CoRea1860+ shows strong agreement with ERA-20C, CERA-20C, and 20CRv3 in both boreal winter (DJF) and summer (JJA) (Figure 13). Notably, ACCs with 20CRv3 are higher in JJA than in DJF, whereas ACCs with CERA-20C and ERA-20C are higher in DJF than in JJA. In the mid-latitudes, ACCs in the Southern Hemisphere and North America are higher in DJF compared to JJA, while ACCs in Eurasia are lower in DJF than in JJA. As for PRECP, the tropical Pacific teleconnection signal over North and South America in DJF is evident (Leathers et al., 1991), while land regions far from the ocean and upstream of ENSO teleconnections, such as Eurasia, have lower ACCs. In the Arctic, positive ACCs are observed across different comparison datasets, except for ACCs with 20CRv3 in JJA. In the Antarctic, prominent ACCs are evident primarily in the Amundsen and Bellingshausen Seas, linked to ENSO teleconnections.

## 5 Conclusions

In this study, we introduce CoRea1860+, a 30-member coupled climate reanalysis spanning from 1860 to the present. This reanalysis was developed using the NorCPM and assimilating SST observations. NorCPM combines the fully coupled Earth System Model NorESM with the EnKF assimilation method. Given that SST was the primary source of instrumental oceanic measurements prior to the 1950s, CoRea1860+ focuses exclusively on assimilating SST data, omitting ocean subsurface observations which only became widely available in recent decades. Moreover, CoRea1860+ was generated in a single continuous stream, ensuring temporal consistency and using an anomaly-field assimilation framework. These methodological choices enhance the continuity of the reanalysis. However, CoRea1860+ does not assimilate atmospheric observations, which limits its

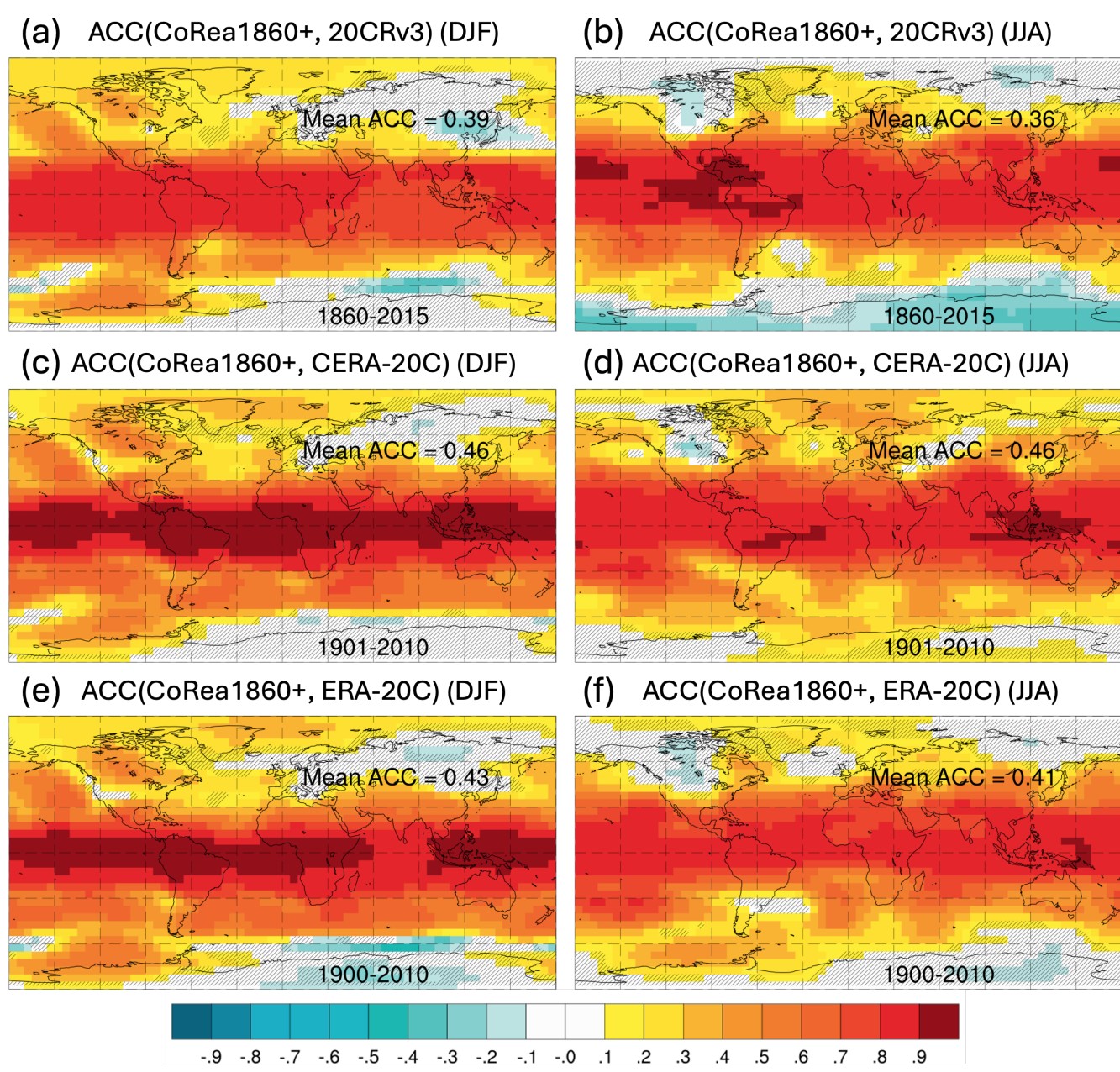

**Figure 13.** ACC of Z500 in DJF or JJA of CoRea1860+ against 20CRv3, CERA-20C and ERA-20C. Grids that fail the significance test are marked with a slash. The period used to compute ACC corresponds to the overlapping period between CoRea1860+ and the specific reference dataset and is shown in Antarctica. The spatially averaged ACC is shown in Eurasia.

ability to fully represent atmospheric variability. The atmospheric component has a horizontal resolution of approximately 2°, while the ocean and sea ice components are resolved at about 1°, both of which are relatively coarse. In addition, CoRea1860+ employs a low-top atmospheric model with limited vertical resolution in the lower stratosphere, which restricts its ability to simulate stratospheric dynamics and stratosphere–troposphere interactions. The use of an anomaly-field assimilation framework also means that large model biases persist. We plan to address these limitations in future versions of CoRea1860+.

The CoRea1860+ reanalysis system demonstrates stability and reliability, particularly during observation-rich periods. To evaluate its reliability, we used assimilation diagnostics, including the temporal evolution of global bias, observation error, background error, and RMSE for the assimilated variable (i.e., SST). Our analysis reveals that the reanalysis system is stable, with no significant bias drift throughout the entire reanalysis period. In the satellite era, when SST observations are characterized by high global coverage and quality, the system performs reliably. However, before this period, the ensemble is slightly overdispersive. Also, the gridded SST product relies on interpolating and extrapolating sparse in situ observations, posing challenges in assessing the reliability of the reanalysis during earlier times.

CoRea1860+, which assimilates SST data, captures the temporal variability of the OHC across different regions and the AMOC. This is primarily due to two factors. First, assimilating SST allows the system to update the interior ocean through ensemble-derived covariances (Counillon et al., 2016; Wang et al., 2022). Second, during model integration, the upper-ocean updates—strongly correlated with SST—are gradually propagated into the deeper ocean. The mean state of the AMOC in CoRea1860+ remains close to that of the free model simulation without DA (Table 2 and Bentsen et al., 2013), as the anomaly-field assimilation framework primarily adjusts anomalies rather than the full fields. For the time evolution of OHC in the upper 300 m, CoRea1860+ captures both the warming trend and slow variability within the range of the reference datasets, including ORA-20C, SODA2.2.4, CHOR, CHORE, and EN4.2.2. Regarding OHC in the upper 2000 m, although significant differences are observed across datasets, CoRea1860+ reproduces the overall warming trend and notable multidecadal variability. For both OHCs in the 0–300 m and 0–2000 m, CoRea1860+ aligns more closely with the comparison datasets in the oceans [60°S, 60°N] and the Arctic Ocean than in the Southern Ocean, particularly after the 1950s. In terms of local OHC variability, despite not assimilating hydrographic profile data, CoRea1860+ shows strong agreement with the comparison datasets. The global mean ACCs with these datasets range from $0.27$ to $0.47$ for OHC in the upper 300 m and from $0.19$ to $0.35$ for OHC in the upper 300 m. However, exceptions are observed in regions such as the Arctic Ocean, Southern Ocean, and tropical Atlantic, where observations are very sparse or models have large deficiencies (Richter, 2015; Counillon et al., 2021). The AMOC in CoRea1860+ exhibits a slight declining trend over the past 160 years, consistent with findings from other CMIP5 models (Cheng et al., 2013) and AMOC reconstructions based on proxy data (Rahmstorf et al., 2015; Caesar et al., 2018). Regarding AMOC variability, CoRea1860+ differs notably from the other ocean datasets (that do not capture the decreasing trend nor simulate multidecadal variability) but demonstrates strong alignment with the RAPID array observations, which have been available since 2004, and other estimates from oceanographic data suggesting stronger AMOC in the 1950s and 1990s, weaker AMOC in the 1970s (e.g., Rahmstorf et al., 2015). This agreement provides additional confidence in the reliability of CoRea1860+'s AMOC reconstruction over the whole period.

CoRea1860+ demonstrates variability in sea ice concentration and extent (SIE) for both the Arctic and Antarctic regions, as the sea ice state can be updated through SST assimilation (Bethke et al., 2021) and subsequent model integration (Wang et al., 2019). In the Arctic, CoRea1860+ captures the long-term declining trend in SIE and exhibits significant variability, consistent with SIBT1850, IAPICE1, and HadISST2.2. It reproduces the multi-decadal variations in the IAPICE1 dataset. However, it underestimates the rate of decline during the satellite era and shows an offset in the 2010s, which is attributed to the transition in the assimilated dataset from HadISST2.1 to OISSTV2. Regarding sea ice concentration in March and September, CoRea1860+ aligns with SIBT1850 and IAPICE1 better in marginal ice regions than in ice pack regions, with seasonal variations, and agrees with HadISST2.2 primarily in the Central Arctic due to the use of climatological data in HadISST2.2 to fill gaps where in situ observations are unavailable. In the Antarctic, CoRea1860+ exhibits behavior similar to HadISST2.2, including a pronounced decline in SIE from 1940 to 1980, followed by a slight increase in subsequent decades. The 1940–1980 decline is consistent with findings from previous studies (e.g., Fogt et al., 2022; Dalaiden et al., 2023; Goosse et al., 2024; Divine et al., 2024), while the slight increase in SIE after 1980 has been well observed in satellite records. For the variability of ice concentration, CoRea1860+ agrees better with HadISST2.2 in March in the Kong Håkon and Ross Seas, and in September in the Atlantic and Pacific sectors.

CoRea1860+, which assimilates SST observations into its ocean and sea ice components and does not use atmospheric observations, represents atmospheric variability to some extent mostly due to robust air-sea and air-sea ice interactions. For surface air temperature, the global mean ACCs between CoRea1860+ and ERA-20C, CERA-20C, and 20CRv3 are approximately 0.56, 0.65, and 0.52 respectively, primarily due to their reliance on similar SST datasets. The agreement is notably higher over oceans than over land, particularly in tropical ocean regions where stronger air-sea interactions dominate. However, agreements are weaker in boreal summer (JJA) than boreal winter (DJF) in North America, likely due to weaker large-scale atmospheric teleconnections originating from the tropics, and in North Asia. For precipitation, CoRea1860+ shows global mean ACCs with the comparison datasets ranging from 0.20 to 0.25 in boreal winter, and from 0.19 to 0.22 in boreal summer. It captures precipitation variability more effectively in the tropics, regions influenced by western boundary currents, and the Southern Ocean. CoRea1860+ exhibits global mean ACCs with these datasets ranging from 0.20 to 0.35 for sea level pressure and from 0.36 to 0.46 for 500 hPa geopotential height, largely driven by the influence of ENSO and its teleconnections. The ACC value is particularly high in the tropics but tends to be low over land regions farther from the ocean. CoRea1860+ does not capture the year-to-year fluctuations of the NAO in phase with the other reanalyses. Nevertheless, it shows a weak NAO's decadal variability, suggesting a potential oceanic influence on longer timescales.

While the present study focuses on describing and evaluating the CoRea1860+ dataset, its extended temporal coverage is well-suited for studying long-term climate variability and the slower modes of the climate system over the historical period. For instance, we have been contributing a prospective paper on AMOC variability and predictability. Furthermore, the direct constraint of the ocean component in CoRea1860+ enhances its suitability for investigating the ocean's role as a primary driver of interactions within the climate system. We have been investigating whether the ocean drives the decline in Antarctic SIE from 1940 to 1980. The relatively large ensemble of 30 members offers possibilities to study teleconnections over the entire historical period. CoRea1860+ can also be used to initialize climate predictions over much longer periods than normally

considered (e.g., 1960-present in CMIP6 Decadal Climate Prediction Project, Boer et al., 2016), and can provide more reliable estimates of multi-annual prediction skill. We have performed a set of decadal hindcasts starting each year from 1873 to 2020

initialized by CoRea1860+ and have been investigating the modulation of decadal prediction skill.

## 6    Data availability

The CoRea1860+ dataset is available at https://doi.org/10.11582/2025.00009 (Wang and Counillon, 2025).

*Author contributions.*    YW, FC, NK and PL conceived the idea. YW and FC designed and conducted the reanalysis experiment with support from PC. YW and FC also evaluated the dataset. YW wrote the manuscript with contributions from FC, LS, NK, PL, and ED. MK uploaded

and published the dataset in the NIRD Research Data Archive. All authors contributed to the final manuscript.

*Competing interests.*    The authors declare that no competing interests are present.

*Acknowledgements.*    This study was supported by the Research Council of Norway (Grant Nos. 301396 and 350390), the Trond Mohn Foundation under project number BFS2018TMT01, and the Bjerknes Centre for Climate Research (BCCR) – Centre of Climate Dynamics (SKD) Strategic Project PARCIM (Proxy Assimilation for Reconstructing Climate and Improving Model). We thank Sigma2 for computing

and storage resources (nn9039k and NS9039K).

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
