# Peer review of "An ensemble-based coupled reanalysis of the climate from 1860 to the present (CoRea1860+)"

_Earth System Science Data, 2025_

## Author Comment (AC1)

**Authors' Response to Reviews of**

**An ensemble-based coupled reanalysis of the climate from 1860 to the present (CoRea1860+)**

Yiguo Wang, François Counillon, Lea Svendsen, Ping-Gin Chiu, Noel Keenlyside, Patrick Laloyaux, Mariko Koseki, and Eric de Boisseson Earth System Science Data, Manuscript ID: essd-2025-127

RC: *Reviewers' Comment*, AR: Authors' Response,

Manuscript Text

**1. General comments:**

RC: The authors present an analysis of their comprehensive reanalysis CoRea1860+, which is driven by SST observational products and covering almost the whole of the historical time period (in a climate science sense) from 1860 until the present. The manuscript in its present form is a rock solid data description manuscript, which should be published in ESSD. The corresponding dataset under https://doi.org/10.11582/2025.00009 is accessible to me.

I do have some comments below to improve or clarify some of the authors methodology and statements. I pledge the authors to consider them.

AR: We sincerely appreciate the reviewer's positive evaluation of our manuscript, and we thank the reviewer for the thoughtful comments and constructive suggestions to improve the manuscript. We have carefully addressed the reviewer's concerns and revised the manuscript accordingly.

Below, we present each comment from the reviewer (Reviewer Comment, **RC**) followed by our response (Authors' Response, AR) and, where applicable, the corresponding changes made to the manuscript (highlighted within the black box  $\Box$ ).

Please find our detailed, point-by-point responses below. We again thank the reviewer for his/her valuable time and effort in reviewing our manuscript.

RC: abstract l.15f "Furthermore, CoRea1860+ aligns well with the other datasets for surface air temperature, precipitation, sea level pressure, and 500 hPa geopotential height, especially in the tropics where air-sea interactions are most pronounced."

This statement appears a bit too optimistic, given the coarse resolution of the stratosphere and the missing atmosphere observations. In particular the authors' results concerning 500 hPa geopotential height, as good as they may be given the fact that only "SST" was assimilated, could be well improved with just some "truly" coupled reanalysis incorporating the few atmosphere observations, which are available prior to 1950.

AR: We thank the reviewer for this comment and agree that the statement is too optimistic. Indeed, the reference atmospheric reanalyses used in this study (i.e., ERA-20C, CERA-20C, and 20CRv3) have assimilated the atmospheric observations, which are sparse in time and space before the 1950s, and achieve a tighter reconstruction of atmospheric variability. We dampened our statement (L15-17) as follows:

Furthermore, CoRea1860+ agrees with the reference atmospheric datasets to some extent for surface air temperature, precipitation, sea level pressure, and 500 hPa geopotential height, especially in the tropics where air-sea interactions are most pronounced.

- RC: 1.129f "This version incorporates emissions and new aerosol-cloud interaction schemes..." 1.132f "the model employs the version 7 coupler" Correction to the previous comment: Please do not consider any comments for 1.129f and 1.132f.
- AR: As the reviewer asked in the latest version of the review, no change was made to these lines.
- RC: 1.135 "The atmosphere component consists of 26 hybrid sigma-pressure levels, extending up to 3 hPa." Does it properly resolve the stratosphere for a proper QBO?
- AR: The atmospheric component of the used version of NorCPM is a low-top atmospheric model with a few layers in the lower stratosphere. We cannot expect it to capture stratospheric dynamics and stratospheric-tropospheric interactions, such as QBO.
- RC: 1.138 "The NorESM used in this study is forced by CMIP5 historical forcings before 2005," Why not CMIP6 external forcings? Expected differences from CMIP6 forcing?
- AR: We appreciate the reviewer's questions. For clarity, we added the following statements to the manuscript (L146-152):

We have extensive experience with this NorESM version (e.g., Counillon et al., 2014, 2016). Most of our DA system has been specifically tuned for this setup (e.g., Wang et al., 2016, 2017; Kimmritz et al., 2018; Wang et al., 2022). While CMIP6 forcings represent the latest update, their implementation in NorCPM has introduced issues. For instance, Passos et al. (2023) have reported that the CMIP6-forced version of NorCPM suffers from artificial bugs in the land use updates, leading to unrealistic land–cryosphere cooling trends. These artefacts result in slightly larger global mean biases and RMSEs compared to the CMIP5-forced version. Further details can be found in the Supplement of Bethke et al. (2021). Therefore, we opted for producing the reanalysis with the robustly tested version of the system that uses the CMIP5 forcing configuration.

- RC: 1.148ff "Since 2011, the monthly SST data from the Optimum Interpolation SST version 2 (OISSTV2, Reynolds et al., 2002) are assimilated, because HadISST2.1 is only available until 2010." Why not HadISST, which goes through until present?
- AR: It is because HadISST does not provide uncertainties, which are one key quantity for data assimilation Evensen (2009). OISSTV2 provides weekly SST and weekly observation error variance, in addition to monthly SST. In our assimilation system, the observation error variance of the monthly data is estimated as the harmonic mean of weekly error variances provided by OISSTV2 (Bethke et al., 2021).

**RC: 1.161f "The SST data in the regions covered by sea ice are not assimilated. These regions are identified using the sea ice mask in HadISST2.1 or OISSTV2." How does the model's observed sea ice extend comply with the observed sea ice? Biases?**

AR: The version of NorESM used in this study underestimates the sea ice extent during winter and overestimates it during summer in the Northern Hemisphere. It overestimates both winter and summer extent in the Southern Hemisphere. Further details can be found in Figures 4 and 11 of Bentsen et al. (2013).

**RC: 1.186f "The climatology reference period is 1950-2009 covering a long observation-rich period." This time period is strongly influenced by climate change signal. Would other time periods be a better choice? I presume that only the satellite era represents an "observation-rich period" in terms of global SST coverage. Please comment on that.**

AR: We agree with the reviewer that flagging the period 1950–2009 as an observation-rich period was an overstatement. Only the satellite era can be considered an observation-rich period for SST data. The climatology period needs to be as long as possible to damp out the contribution of internal variability in the estimate of the climatology in the observational record (considering that the true climate is a single realization). We cannot use data beyond 2010 because HadISST2 is not available. Before 1950, the SST observation was strongly undersampled and thus inaccurate in some regions (e.g., the Southern Ocean). The period 1950–2010 appears as a long period when monthly SST estimates are relatively accurate for estimating monthly climatology. We have restricted it to 2009 to get a 60-year climatology period. In addition, Bethke et al. (2021) have tested different climatology periods within NorCPM, e.g., 1980-2010 and 1950-2010, and have found that the different climatological periods lead to similar multiyear AMOC variations. There were some discrepancies on the long-term trends of the North Atlantic subpolar gyre circulation and AMOC at high latitudes, but these differences were mostly related to spurious updates of the deep water masses that got corrected by using the vertical localization technique (Wang et al., 2022). For clarity, we revised the relevant statement (L196-198) as follows:

Referring to Bethke et al. (2021), we define 1950-2009 when monthly SST estimates are relatively accurate for estimating monthly climatology as the climatology reference period to avoid subsampling the internal variability of the climate.

**RC: Further below, the anomalies of OHC and AMOC are against 1950-2010.**

AR: To ensure consistency with the climatology period used in the anomaly-field data assimilation, we applied the 1950–2009 period for Figures 2 (OHC), 4 (AMOC), and 8 (SAT) in the revised manuscript. Please note that using either 1950–2009 or 1950–2010 as the climatology period results in only negligible differences in these figures, with no impact on the overall conclusions.

**RC: 1.243 "RAPID" Please add reference Moat et al. (2024) in this paragraph.**

AR: As suggested, we added the reference of RAPID to the manuscript (L255-256):

**RAPID** (Moat et al., 2024) makes use of arrays of moorings to monitor the variability of the meridional overturning circulation at 26° N in the Atlantic and has sustained the observations since 2004.

- RC: 1.299, Figure 1 Please make the legend smaller and moved up a bit to reveal all of the time series.
- AR: We thank the reviewer for the suggestion. We revised Figure 1 of the manuscript as Figure R1.
- RC: 1.337ff 4.2.1 Ocean heat content Within this subsection, it is a bit confusing to jump forth and back several times between 0-300m OHC and 0-2000m OHC. Please consider to re-order the paragraphs so that 0-300m comes first and 0-2000m OHC second.
- AR: As suggested, we modified the structure of Section 4.2.1. Please refer to the revised version of the manuscript (L356-398).
- **RC:** 1.384 Figure 3 Please consider to add the corresponding time periods as labels in plot, e.g. over Antarctica.

Figure R1: Global assimilation diagnostics for the assimilated variable – monthly SST anomalies: bias (blue line), background error (orange line), observation error (green line), total error (red line), and RMSE (purple line).

Table T1: Temporal mean and standard deviation of the maximum AMOC transport at 26° N over 2005–2021 for RAPID and 1950–2009 for the other datasets.

| Dataset    | Mean (Sv) | Standard deviation (Sv) |
|------------|-----------|-------------------------|
| ORA-20C    | 11.4      | 1.7                     |
| SODA2.2.4  | 16.2      | 2.5                     |
| CHOR       | 15.6      | 2.0                     |
| CHORE      | 16.0      | 2.0                     |
| RAPID      | 16.7      | 1.4                     |
| CoRea1860+ | 32.0      | 1.8                     |

AR: We revised Figure 3 of the manuscript as Figure R2.

**RC: 1.405 4.2.2 Atlantic meridional overturning circulation Please add a statement on absolute values for the AMOC, or the mean biases of the systems, in particular in context with Figure 4.**

AR: We added a table on the temporal mean and standard deviation of the AMOC (Table T1) and some descriptions into the manuscript (L433-436):

While CoRea1860+ exhibits a standard deviation in maximum AMOC transport at 26° N within the range of that of the reference datasets over 1950–2009 (except for RAPID over 2005–2021), its mean AMOC strength at 26° N is notably higher, with a time-mean of 32.0 Sv (Bentsen et al., 2013), exceeding that of the other datasets (Table 2).

---

## Author Comment (AC2)

**Authors' Response to Reviews of**

**An ensemble-based coupled reanalysis of the climate from 1860 to the present (CoRea1860+)**

Yiguo Wang, François Counillon, Lea Svendsen, Ping-Gin Chiu, Noel Keenlyside, Patrick Laloyaux, Mariko Koseki, and Eric de Boisseson Earth System Science Data, Manuscript ID: essd-2025-127

RC: Reviewers' Comment, AR: Authors' Response,

- RC: This paper presents a new and interesting contribution to the ocean reanalysis landscape with the development of CoRea1860+, a long-term coupled ocean reanalysis that assimilates only sea surface temperature (SST) observations. By avoiding the assimilation of subsurface data, which is more sparse in space and time, the product is designed to maximize temporal consistency, making it particularly well suited for studies of climate variability on decadal and longer timescales. The manuscript provides a clear and thorough evaluation of CoRea1860+, demonstrating how it compares with both existing ocean reanalyses and observational datasets. The approach is novel and complementary to other reanalyses that rely on more comprehensive data assimilation strategies.
- AR: We sincerely appreciate the reviewer's positive evaluation of our manuscript as a clear and thorough description of the dataset available at https://doi.org/10.11582/2025.00009, and we thank the reviewer for the thoughtful comments and constructive suggestions to improve the manuscript. We have carefully addressed the reviewer's concerns and revised the manuscript accordingly.

Below, we present each comment from the reviewer (Reviewer Comment, **RC**) followed by our response (Authors' Response, AR) and, where applicable, the corresponding changes made to the manuscript (highlighted within the black box  $\Box$ ).

Please find our detailed, point-by-point responses below. Again, we thank the reviewer for his/her valuable time and effort in reviewing our manuscript.

- RC: I recommend minor revision, primarily to encourage the authors to expand their discussion and analysis in a few areas where the strengths of CoRea1860+ could be further leveraged—for example, by using its extended temporal coverage to explore decadally paced modes of variability or by more explicitly investigating the ocean-forced atmospheric variability. These two aspects are already well acknowledged in the paper, but could benefit from deeper exploration to further highlight the utility of the dataset.
- AR: We thank the reviewer for this thoughtful and constructive suggestion. We fully agree that the extended temporal coverage of CoRea1860+ and its explicit representation of ocean–atmosphere interactions present valuable opportunities for studying decadal variability and ocean-forced atmospheric responses. As noted by the reviewer, these aspects are acknowledged and partially explored in the manuscript. To better align with the reviewer's recommendation, we added a new analysis of the North Atlantic Oscillation (NAO, Figure R1), a key atmospheric mode of climate variability, which offers further insight into the capability of CoRea1860+ to represent ocean-forced atmospheric decadal variability. The new content was included in Section 4.4.3 (L580-590) and the conclusions section (L668-670) as follows:

The North Atlantic Oscillation (NAO) is a predominantly atmospheric mode of climate variability and represents one of the most prominent patterns of climate fluctuations over the North Atlantic and

Figure R1: (a) Winter (December-March) station-based index of the NAO. (b) 10-year running average of the winter NAO index.

surrounding regions (e.g., Hurrell, 1995). Following the definition in Hurrell (1995), we calculate the winter (December–March) NAO index as the difference between normalized SLPs at Lisbon and Stykkisholmur (Figure 12). The NAO indices from ERA-20C, CERA-20C, and 20CRv3 show strong agreement, reflecting the constraint provided by the assimilation of surface pressure observations (Figure 12a). In contrast, CoRea1860+, which assimilates only SST observations, exhibits weak interannual variability in the ensemble mean and a relatively large ensemble spread. The correlations between the NAO indices from CoRea1860+ and those from the reference reanalyses are all below 0.16. This indicates that SST assimilation alone is not sufficient to synchronize the NAO's year-to-year fluctuations with those in the other reanalyses. Nevertheless, CoRea1860+ shows the NAO's decadal variability (Figure 12b), suggesting a potential oceanic influence on longer timescales. This highlights an area of interest for future investigation, particularly in exploring how ocean-atmosphere coupling may contribute to low-frequency NAO variability.

CoRea1860+ does not capture the year-to-year fluctuations of the NAO in phase with the other reanalyses. Nevertheless, it shows a weak NAO's decadal variability, suggesting a potential oceanic influence on longer timescales.

A more in-depth exploration of decadal variability and ocean-forced atmospheric dynamics is indeed of high scientific interest and will be a key focus of future studies. To briefly clarify this intent, we added a forward-looking paragraph to the conclusions section of the manuscript (L671-680):

While the present study focuses on describing and evaluating the CoRea1860+ dataset, its extended temporal coverage is well-suited for studying long-term climate variability and the slower modes of the climate system over the historical period. For instance, we have been contributing a prospective paper on AMOC variability and predictability. Furthermore, the direct constraint of the ocean component in CoRea1860+ enhances its suitability for investigating the ocean's role as a primary driver of interactions within the climate system. We have been investigating whether the ocean drives the decline in Antarctic SIE from 1940 to 1980. The relatively large ensemble of 30 members offers possibilities to study teleconnections over the entire historical period. CoRea1860+ can also be used to initialize climate predictions over much longer periods than normally considered (e.g., 1960-present in CMIP6 Decadal Climate Prediction Project, Boer et al., 2016), and can provide more reliable estimates of multi-annual

prediction skill. We have performed a set of decadal hindcasts starting each year from 1873 to 2020 initialized by CoRea1860+ and have been investigating the modulation of decadal prediction skill.

**1. Specific comments**

**RC: Line 10: Change "areas" to "aspects".**

AR: We revised the text (L10) as:

It then provides a comprehensive evaluation of the reanalysis across four key aspects

**RC: Line 27: Change "Retrospective analysis... is" to "Retrospective analyses... are".**

AR: Done. The revised text (L28-30) is as follows:

Retrospective analyses (i.e., reanalyses, Kalnay et al., 1996; Wang et al., 2023) are a comprehensive four-dimensional reconstruction of the historical climate system achieved by combining observational data (i.e., observations) with a numerical physical model through data assimilation (DA, Evensen, 2003; Carrassi et al., 2018; Penny et al., 2017).

**RC: Line 66: Rephrase "and the discard of the first two years of each streamline" to "from which the first two years are discarded".**

AR: Thanks for the suggestion. We revised the text (L65-68) as

its production process involved the parallel generation of 14 ten-year production streams (initialized from the uncoupled ERA-20C and ORA-20C reanalyses) from which the first two years are discarded to produce the final climate reconstruction for the period 1901–2010, leading to discontinuities in the ocean variables (see Figure 10 in Laloyaux et al., 2018).

**RC: Lines 76-78: It is not very clear what distinguishes semi-coupled from fully-coupled assimilation. Could you elaborate a bit more?**

**AR: Sorry for the confusion. For clarity, we modified the text (L76-84) as follows:**

Coupled DA can be classified into two types: semi-coupled DA and fully-coupled DA. In semi-coupled DA, observations are assimilated into their respective components (Counillon et al., 2016; Kimmritz et al., 2019). For instance, when assimilating atmospheric and oceanic data in semi-coupled DA, the atmospheric data are only used to update the atmosphere component and the oceanic data are solely used to update the ocean component. Semi-coupled DA still allows for constraints on the other components through the model's coupling of the components, making it valuable for studying specific component interactions within the climate system. In fully-coupled DA, all components are directly constrained by observations (Fujii et al., 2009; Laloyaux et al., 2016). Again, for example, when assimilating atmospheric and oceanic data, the atmospheric data are used to update not only the atmosphere component but also the ocean component and the oceanic data are used to update both the ocean and atmosphere components.

- RC: Lines 87-89: The way it is introduced, it seems that the problems of full-field initialization (e.g. model drift) are specific to uncoupled reanalyses. This is misleading as I would actually expect uncoupled reanalyses to be less subject to model drift due to the anchoring effect of the boundary conditions. Since full-field initialization is also used in coupled reanalyses, I would introduce it as a standalone concept that is opposed to anomaly initialization.
- AR: We agree with the reviewer that the statements were misleading. For clarity, we revised the text (L92-96) as

The full-field assimilation directly uses the actual value of observations and corrects both the model state and variability (de Boisséson et al., 2018; Slivinski et al., 2021). However, this approach can lead to persistent model drift, where the coupled system repeatedly moves away from observations toward its biased state (Carrassi et al., 2014; Weber et al., 2015). This drift introduces inconsistencies in the coupled reanalysis, particularly in unobserved variables and regions such as the deep ocean when observational data are sparse.

**RC: Line 113: Change "reconstruction" to "reconstructions".**

AR: As suggested, the text (L117-119) was rewritten as

NorCPM is a physics-based numerical model (scientific software) developed for performing climate reconstructions (Counillon et al., 2016; Wang et al., 2022) and predictions on different timescales (Wang et al., 2019; Bethke et al., 2021; Nair et al., 2024; Xiu et al., 2025).

**RC: Lines 147-150: Could you comment what impact this discontinuity in the assimilated product is expected to have in your reanalyses? Could you also motivate the choice of this reference dataset and explain in particular why you didn't consider HadISST1 which extends up to present?**

AR: HadISST2.1 has ten realizations of monthly gridded SST. The standard deviation between the ten realizations, which varies with time and space, is designed to reflect its uncertainties - one key quantity for data assimilation (Evensen, 2009). However, HadISST1 does not provide uncertainties. OISSTV2 provides weekly SST and weekly observation error variance, in addition to monthly SST. In our assimilation system, the observation error variance of the monthly data is estimated as the harmonic mean of weekly error variances provided by OISSTV2 (Bethke et al., 2021).

HadISST2.1 is our preferred dataset, but it is only available until 2010. During the preparation of the Decadal Climate Prediction Project (DCPP) experiments for the sixth phase of the Coupled Model Intercomparison Project (CMIP6) (Bethke et al., 2021), we verified through a separate reanalysis covering the period 2006–2010 that the transition from HadISST2 to OISSTv2 does not introduce any noticeable discontinuities in the coupled reanalysis.

**RC: Lines 312-313: I wouldn't say that the total error is stable as it shows a clear decreasing trend over time.**

AR: We agree with the reviewer and revised the text (L324-326):

The total error (red line in Figure 1) is about  $0.6 \,^{\circ}$ C with a slight decreasing trend before 1982 and about  $0.5 \,^{\circ}$ C after 1982. Its shrinkage in 1982 is mainly because of introducing the satellite observations.

RC: Figure 2 and paragraphs in Lines 356-376: Can you motivate the choice of the two polar regions as regions that deserve specific validation? Is it because they have been more poorly observed before the satellite era?

AR: As suggested, we added one paragraph into Section 4.2.1 (L351-355):

We evaluate three ocean regions: the open-water ocean  $[60^{\circ}S, 60^{\circ}N]$ , the Arctic Ocean  $[60^{\circ}N, 90^{\circ}N]$ , and the Southern Ocean  $[60^{\circ}S, 90^{\circ}S]$ . In each region, we assess OHC in both the upper 0–300 m and 0–2000 m layers. The two polar regions are of particular interest, as they are critical for climate studies but were poorly observed before the satellite era. Moreover, SST data in areas covered by sea ice are not assimilated, making it especially valuable to evaluate the system's performance under sea ice conditions.

**RC: Lines 359-360: I don't see why having more data available should traduce in higher variability. Isn't the higher variance in this region with respect to the global ocean explained by the polar amplification phenomenon?**

AR: We agree with the reviewer that the statements were misleading. For clarity, we modified the text (L366-368) as

It should be highlighted that the hydrographic profile data availability at high latitudes is higher. Over the period 1980-2010, EN4.2.2 depicts a higher warming trend in the Arctic than in the open-water ocean ( $[60^{\circ}S, 60^{\circ}N]$ ), which confirms the polar amplification phenomenon.

**RC: Lines 362-363: The apparent good agreement after the 1950s might be artificial, as this is the period that you chose to compute the anomalies. If you choose a different one (the early 20th century or the whole dataset, the periods of agreement/disagreement might be very different.**

AR: Using another period as the climatology period would shift up or down the timeseries of different datasets. But their temporal variability is still the same. As we mostly focus on the temporal variability evaluation in this study, we revised the sentence (L371-372) as follows:

From the 1950s onward, all datasets are in good agreement in temporal variability, while EN4.2.2 shows higher variability amplitude than the other products.

**RC:** Lines 400-401: How much of this good agreement is because of the long-term trends. Have you recomputed the correlations with detrended timeseries?**

AR: In the manuscript, we have computed correlations for the anomalies including the trends (Figure R2, i.e., Figure 3 in the manuscript). To address the reviewer's comment, we computed correlations for detrended anomalies (Figure R3). Comparing Figure R3 to Figure R2 demonstrates that the agreement between CoRea1860+ and the reference datasets is mostly due to the synchronization of the yearly variability. Long-term trends do contribute to the ACC maps, which are relatively weak. We added the following statement to the manuscript (L426-427):

Note that interannual variability in OHC contributes much more to the computation of ACCs than the long-term trend (not shown).

RC: Section 4.2.2: It is important to point out that CoRea1860+ cannot describe the observed Ekman-driven component, as it doesn't include any wind forcing. That might explain some of the year-to-year discrepancies with the other reanalyses. It might be worth repeating the comparison but removing first the Ekman component (e.g. as in Baehr et al 2004).